# Mechanism of selective recognition of Lys48-linked polyubiquitin by macrocyclic peptide inhibitors of proteasomal degradation

Betsegaw Lemma[1], Di Zhang[2], Ganga B. Vamisetti[3], Bryan G. Wentz [1], Hiroaki Suga [4], Ashraf Brik [3] ✉, Jacek Lubkowski[2] ✉ & David Fushman [1] ✉

Post-translational modification of proteins with polyubiquitin chains is a critical cellular signaling mechanism in eukaryotes with implications in various cellular states and processes. Unregulated ubiquitin-mediated protein degradation can be detrimental to cellular homeostasis, causing numerous diseases including cancers. Recently, macrocyclic peptides were developed that selectively target long Lysine-48-linked polyubiquitin chains (tetra-ubiquitin) to inhibit ubiquitin-proteasome system, leading to attenuation of tumor growth in vivo. However, structural determinants of the chain length and linkage selectivity by these cyclic peptides remained unclear. Here, we uncover the mechanism underlying cyclic peptide's affinity and binding selectivity by combining X-ray crystallography, solution NMR, and biochemical studies. We found that the peptide engages three consecutive ubiquitins that form a ring around the peptide and determined requirements for preferential selection of a specific trimer moiety in longer polyubiquitin chains. The structural insights gained from this work will guide the development of next-generation cyclic peptides with enhanced anti-cancer activity.

Ubiquitin (Ub) is a small, 76-residue protein that acts as a post-translational modification of proteins in eukaryotic cells, resulting in diverse signaling pathways[1-4]. Ubiquitination of specific substrates affects a myriad of vital cellular processes including regulated protein turnover, cellular division and differentiation, as well as DNA damage repair[1,4,5]. The diversity of signaling outcomes reflects Ub's ability to form polymeric (polyUb) chains through a covalent linkage between the C terminus of one Ub and any of the seven lysines or the N terminus of another Ub[6]. In particular, polyUb chains linked via K48 have been extensively studied in their function as the primary signal targeting proteins for proteasomal degradation.[7]

Multiple studies have demonstrated that cancer cells utilize the ubiquitin-proteasome system (UPS)[8] to remove critical proteins, which leads to uncontrolled growth and evasion of apoptosis[9,10]. Thus, the UPS pathway which involves several oncogenes is an exceptional target for development of cancer therapeutics[11-13]. In fact, several small molecules which act as proteasome inhibitors have been either approved or are in the process of approval due to their success as anticancer therapeutics[14-18]. Nevertheless, the limits in the types of cancers that respond to these inhibitors and the emergence of resistance underscore the need for better or alternative therapeutics[16,19,20]. On the other hand, the PROTAC approach has emerged as strategy to eliminate oncogenic proteins by designing a bifunctional molecule that recruits a Ub ligase and the targeted proteins for ubiquitination followed by their removal by the proteasome[21]. Ub's exceptionally high sequence conservation among eukaryotes and low tolerance of cells to

[1]Center for Biomolecular Structure and Organization, Department of Chemistry and Biochemistry, University of Maryland, College Park, MD 20742, USA. [2]Center for Structural Biology, Center for Cancer Research, National Cancer Institute, Frederick, MD 21702, USA. [3]Schulich Faculty of Chemistry, Technion – Israel Institute of Technology, Haifa 3200008, Israel. [4]Department of Chemistry, Graduate School of Science, The University of Tokyo, Bunkyo-ku, 7-3-1 Hongo, Bunkyo, Tokyo 113-0033, Japan. ✉e-mail: abrik@technion.ac.il; jacek.lubkowski2@nih.gov; fushman@umd.edu

Ub mutations[22,23] make it a potentially robust therapeutic target. The discovery of ubistatins, small molecules that impair UPS by directly binding to polyUb and blocking its recognition by proteasomal receptors and shuttles, as well as ubiquitination and deubiquitination machineries, brought to the forefront the degradation signal itself as a valid target for therapeutic applications[24,25]. However, in this endeavor, the extent of cellular processes that rely on polyUb chains of various linkage types demands specificity of binding to chains of the right linkage type as well as the proper length[26]. In particular, the focus has been on K48-linked tetra-Ub (Ub$_4$) and longer chains shown to be an efficient proteasome-targeting signal[27], although recent evidence suggests that shorter chains as well as branched chains containing more than one linkage type have some role in this function as well[28–30].

As such, Nawatha et al. had developed de novo cyclic peptides capable of selectively binding K48-linked tetra-Ub chains with nanomolar affinities and protecting them from deubiquitinases (DUBs) and the proteasome[11]. Building on this discovery, Rogers et al. further optimized the peptide requirements and developed highly non-proteinogenic cyclic peptides with improved inhibitory activity that caused accumulation of Ub-conjugates and induced apoptosis in live cells and attenuated tumor growth in vivo[31]. However, how these relatively short, 12 amino-acid cyclic peptides, that are substantially smaller in size compared even to a single Ub unit achieve selective recognition of Ub tetramers remained unclear.

Our preliminary characterization of binding of one of these macrocyclic peptides, termed Ub4a (Fig. 1), to K48-linked polyUb chains using NMR spectroscopy[31] suggested that the peptide primarily interacts with the "canonical" hydrophobic surface patch residues (L8, I44 and V70) in Ub that are essential for proteasomal recognition of polyUb chains[32]. Intriguingly, these NMR and DUB assays revealed that the cyclic peptide selectively engages only three out of the four Ub units in tetra-Ub, located specifically at the proximal (bearing unanchored C terminus) end of the chain. Here we use a combination of X-ray crystallography, solution NMR spectroscopy, and biochemical/DUB assays to uncover the structural mechanism by which the peptide selectively binds to K48-linked tri- and tetra-Ub chains. These studies aim at understanding (1) how such a short cyclic peptide achieves selectivity toward long polyUb chains and (2) how out of the four Ub units in tetra-Ub the peptide preferentially selects specific three Ub units. Our results show that K48-linked tri-Ub wraps around the peptide in a ring-like arrangement, with the main interactions occurring inside the central hole lined with the hydrophobic surface patch residues of all three consecutive Ub units in the chain, and reveal the role of the C-terminal residues of the proximal Ub unit as determinants of where and how the cyclic peptide binds to K48-linked Ub tetramer.

## Results

Throughout this paper we will use specific nomenclature introduced earlier[33] to succinctly convey the type of polyUb chains used for our studies. Each chain is generally written as Ub$_n$ with $n$ being the number of Ub units within the chain. Chains with deletions or extension in the proximal Ub unit (the one bearing free/unanchored C terminus) are written as [Ub]$_n$-Ub$_x$ where $x$ stands for the number of Ub units preceding the proximal Ub (Ub$_x$) and $x$ indicates the last residue in the C-terminal tail of that Ub unit. All chains considered here are linked via K48, thus we will skip the designation of the linkage type. Furthermore, the distal Ub unit (the farthest from the proximal end) in these chains contains a K48R mutation, to prevent chain elongation on the distal end. Inside each chain, Ub units are designated by letters A through C (for Ub$_3$) or A through D (for Ub$_4$) starting from the proximal end. For example, Ub$_3$ can be represented as Ub$_C$-Ub$_B$-Ub$_A$, where Ub$_C$, Ub$_B$ and Ub$_A$ stand for the distal, middle (endo) and proximal Ub units, respectively. Specific residues within each Ub unit are also marked with the respective subscript unit identifier.

## NMR studies reveal cyclic peptide's selectivity for the proximal trimer moiety

Our previous NMR studies identified the proximal Ub trimer moiety (i.e. Ub units A through C) as the main Ub4a-binding element within K48-linked tetra-Ub chain[31], raising the question how Ub4a recognizes the proximal end of tetra-Ub. Specifically, while all three Ub units in tri-Ub exhibited strong NMR signal perturbations upon addition of Ub4a, almost negligible changes were detected in the distal Ub of tetra-Ub (Fig. S1), indicating that this Ub unit does not interact with the peptide[31]. Notably, the $^1$H-$^{15}$N NMR spectra of the proximal Ub in the peptide-bound state of tri- and tetra-Ub were almost identical. A close inspection of the changes in NMR spectra of the proximal Ub (Ub$_A$) in both chains caused by Ub4a binding revealed, in addition to signal perturbations of the hydrophobic patch residues, noticeable shifts in the signals of the C-terminal residues of that Ub unit (Fig. S2). The C-terminal tail (residues L73-R74-G75-G76) of the unbound proximal Ub in polyUb chains exhibits essentially unrestricted backbone flexibility[34,35], therefore the NMR signal shifts of these residues in the presence of Ub4a suggest their close proximity to or even direct contacts with the peptide in the bound state. We therefore hypothesized that the interactions between Ub4a and the C terminus of the proximal Ub play role in the peptide's binding selectivity by directing Ub4a toward the proximal end of the tetra-Ub chain.

In order to test if the C-terminal tail of the proximal Ub plays role in the preferential selection of the proximal trimer moiety by the peptide, here we introduced truncations of the C-terminal tail of the proximal Ub in both tri-Ub and tetra-Ub chains and compared the effect of Ub4a binding on the NMR spectra of these modified chains with that for their un-truncated variants. For this purpose, two types of chains were made containing the following C-terminal truncations in the proximal Ub unit: a) deletion of the last four tail residues (ΔLRGG, referred to here as the entire tail deletion) resulting in the [Ub]$_3$-Ub$_{R72}$ tetramer and [Ub]$_2$-Ub$_{R72}$ trimer; and b) deletion of only the last two residues (ΔGG, partial tail deletion) resulting in the [Ub]$_3$-Ub$_{R74}$ tetramer and [Ub]$_2$-Ub$_{R74}$ trimer (Fig. S3a). These chains were assembled as previously described[31], and the NMR experiments focused on observing the effect of these deletions on the $^1$H-$^{15}$N spectra of $^{15}$N-labeled distal unit (unit D in the tetramer or C in the trimer) of each chain type. Of note, these C-terminal truncations in the proximal Ub had no effect on the NMR spectra of the distal Ub in the unbound Ub$_3$ and Ub$_4$ chains (Fig. S3b).

The Ub chains with the entire tail removed showed NMR spectra in the Ub4a-bound state that were glaringly different from their wild-type counterparts in complex with Ub4a (Fig. 1a, b vs Fig. S1). Interestingly, the titration end-point NMR signals of the $^{15}$N-labeled distal Ub in [Ub]$_3$-Ub$_{R72}$ tetramer showed a remarkable similarity to those of the distal Ub in wild-type Ub$_3$ in the presence of Ub4a (Fig. 1c). This was an indication that the binding mode of Ub4a had changed such that the distal trimer moiety (i.e., Ub units B through D) now became the main peptide-binding site on the tetramer. Furthermore, the spectra of the distal Ub in the [Ub]$_2$-Ub$_{R72}$ trimer (Fig. 1b) exhibited residue-specific signal attenuations but only minor shifts, in contrast with the strong shifts and slow-exchange regime of binding observed for the wild-type trimer (Fig. S1). This is indicative of an intermediate exchange regime and suggests a weaker Ub4a binding to [Ub]$_2$-Ub$_{R72}$. Moreover, somewhat similar to what was observed for [Ub]$_3$-Ub$_{R72}$ vs Ub$_4$, the distal-Ub spectra of Ub4a-bound [Ub]$_2$-Ub$_{R72}$ resembled those of the distal Ub in wild-type Ub$_2$ bound to Ub4a (Fig. 1d). Together, these results indicate that the deletion of the entire C-terminal tail of the proximal Ub abolished the recognition of that Ub unit by the peptide and shifted the site of binding by one Ub unit toward the distal end in both tetra- and tri-Ub chains. (See also below).

By contrast, for the chains with only partial deletion of the C-terminal tail residues (i.e., ΔGG) the spectra of the distal Ub in both [Ub]$_3$-Ub$_{R74}$ and [Ub]$_2$-Ub$_{R74}$ in complex with Ub4a (Fig. 2a, b,

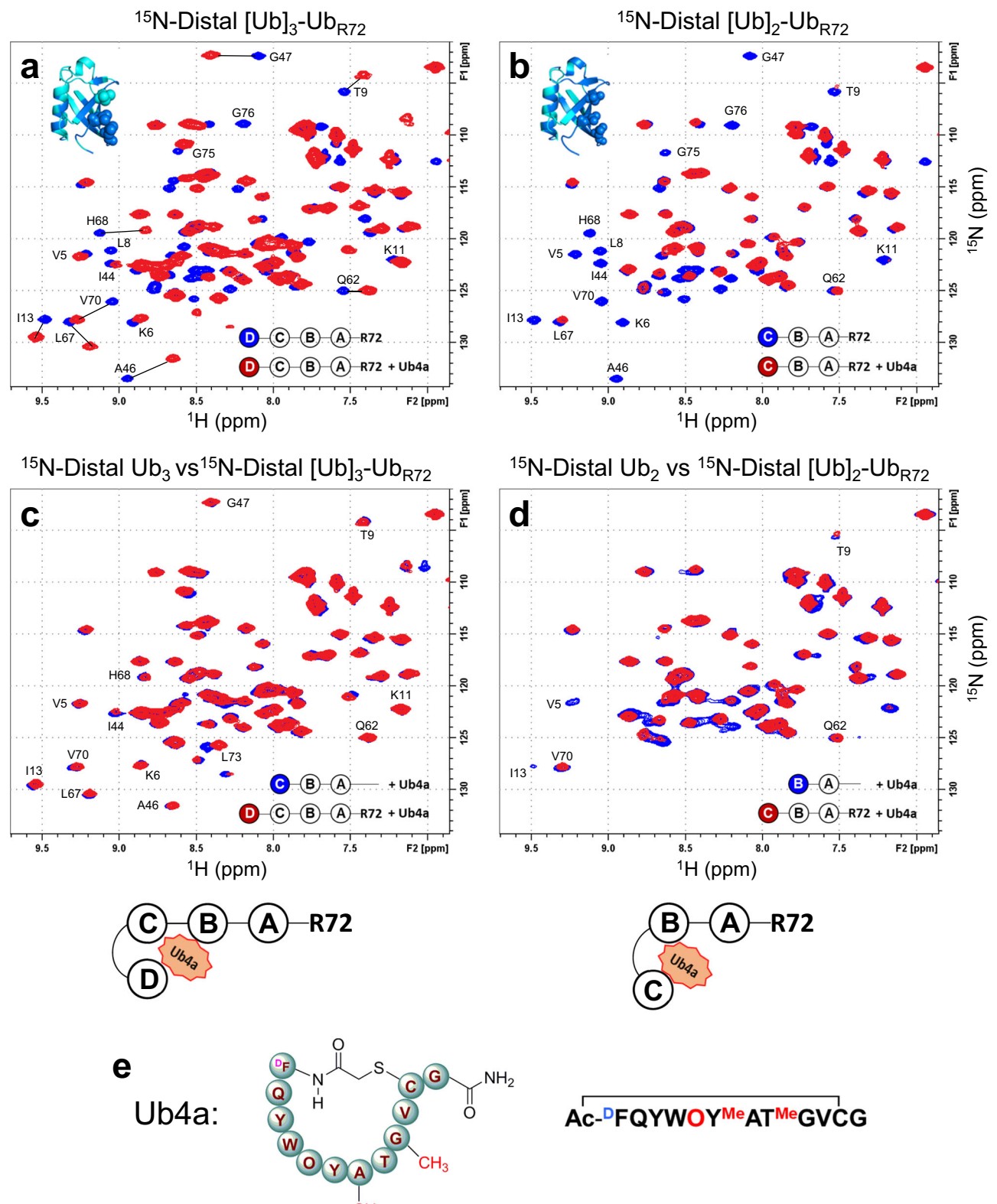

**Fig. 1 | Ub4a binding to the distal Ub of Ub₄ and Ub₃ chains with four C-terminal residues, LRGG, deleted in the proximal Ub (ΔLRGG).** Overlays of ¹H-¹⁵N SOFAST-HMQC NMR spectra of the distal Ub of **a** [Ub]₃-Ub$_{R72}$ free in solution (blue) and upon addition of Ub4a (red); **b** [Ub]₂-Ub$_{R72}$ free in solution (blue) and upon addition of Ub4a (red); **c** wild-type Ub₃ (blue) and [Ub]₃-Ub$_{R72}$ (red) at the endpoint of titration with Ub4a; **d** wild-type Ub₂ (blue) and [Ub]₂-Ub$_{R72}$ (red) at the endpoint of titration with Ub4a. Note the similarities between the blue and red spectra in **c** and also in **d**. Insets in **a** and **b** show Ub structure with perturbed residues colored blue and hydrophobic patch residues shown as spheres. The cartoon drawings below **c** and **d** illustrate the peptide binding arrangement deduced from the NMR data. The peptide:polyUb molar ratio was 1.5:1. **e** Schematic representations of the amino acid composition of the cyclic peptide Ub4a.

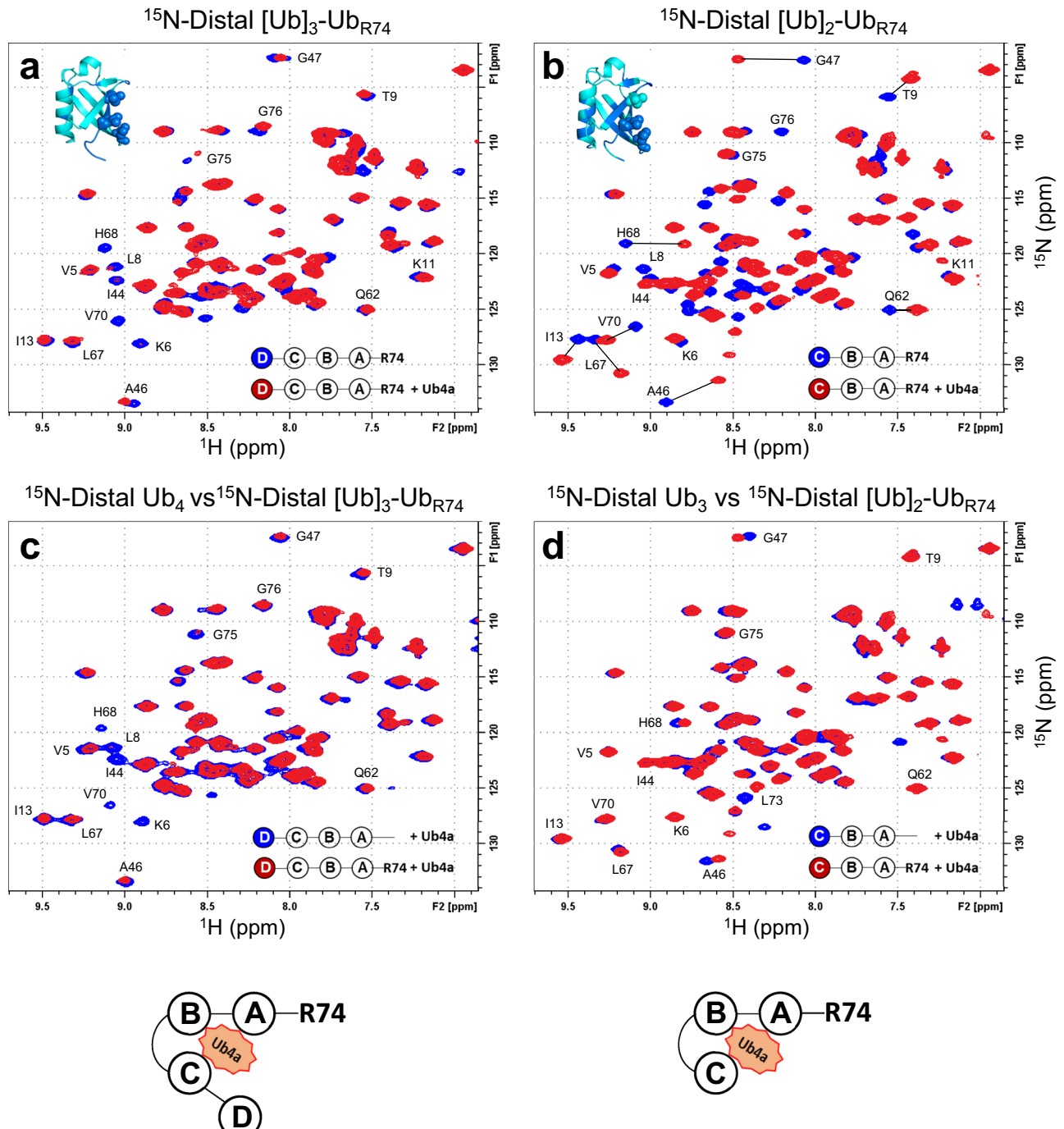

**Fig. 2 | Ub4a binding to the distal Ub of Ub$_4$ and Ub$_3$ chains with two C-terminal residues, G75 and G76, deleted in the proximal Ub (ΔGG).** Overlays of $^1$H-$^{15}$N NMR spectra of the distal Ub of **a** [Ub]$_3$-Ub$_{R74}$ free in solution (blue) and upon addition of Ub4a (red); **b** [Ub]$_2$-Ub$_{R74}$ free in solution (blue) and upon addition of Ub4a (red); **c** wild-type Ub$_4$ (blue) and [Ub]$_3$-Ub$_{R74}$ (red) at the endpoint of titration with Ub4a; **d** wild-type Ub$_3$ (blue) and [Ub]$_2$-Ub$_{R74}$ (red) at the endpoint of titration with Ub4a.

Note the similarities between the blue and red spectra in **c** and in **d**. Insets in **a** and **b** show Ub structure with perturbed residues colored blue and hydrophobic patch residues shown as spheres. The cartoon drawings below **c** and **d** illustrate the peptide binding arrangement deduced from the NMR data. The peptide:polyUb molar ratio was 1.5:1.

respectively) showed strong similarity to their wild-type counterparts with only some slight difference in the directions of a few signal shifts in the trimer (Fig. 2d) and seemingly stronger signal attenuations of the hydrophobic patch residues in the distal Ub of the truncated tetramer (Fig. 2c) compared to wild-type chains. The slight difference in the spectra of the trimers (Fig. 2d) could be due to possible structural rearrangements caused by the tail residue deletions which might also

have some effect on the strength of peptide binding. Overall, these data do not indicate a shift in the binding mode/site that was observed for the chains with entirely deleted C-terminal tail. Note also a strong similarity between the signal shifts of the distal Ub in [Ub]$_2$-Ub$_{R74}$ (Fig. 2b) and in [Ub]$_3$-Ub$_{R72}$ (Fig. 1a), further corroborating the shift in the peptide-binding site on Ub$_4$ upon removal of the entire C-terminal tail.

**Chain disassembly assays show that proximal-Ub tail residues dictate cyclic peptide binding mode and protection from DUBs**

We next utilized deubiquitination reactions to independently verify our NMR findings. Deubiquitinases (DUBs) are a family of (iso)peptidases that regulate Ub-mediated signaling and maintain the presence of free ubiquitin in cells by remodeling or disassembling polyUb chains or removing them from modified proteins[2,36,37]. Specifically, we focused on DUBs known to trim an unanchored polyUb chain from its ends. We hypothesized that strong Ub4a binding to the proximal trimer moiety will render these Ub units sterically inaccessible to and prevent their disassembly by DUBs that preferentially remove Ub monomers from the proximal end of a chain. To test this, we utilized isopeptidase T (IsoT/USP5), a linkage-nonspecific DUB that disassembles unanchored Ub chains from the proximal end[38]. IsoT requires access to intact C-terminal tail (including G76) in order to bind tightly to and cleave the proximal Ub[38,39]. Using pre-assembled $Ub_n$:Ub4a complexes we found that Ub4a effectively protected $Ub_3$, $Ub_4$, $Ub_5$, and longer chains from cleavage by IsoT (Fig. 3a, b), thus confirming the peptide's preferential binding to the proximal trimer moiety in these chains. Noteworthy, Ub4a was inefficient in protecting di-Ub, consistent with the peptide's significantly lower affinity for $Ub_2$ ($K_d > 2\ \mu M$[31]) compared to IsoT's (2 nM[39]).

We then used OTUB1, a K48-linkage specific DUB that has been shown to disassemble unanchored polyUb chains from either end[40]. Our results indicate that Ub4a binding to the proximal trimer moiety protects K48-linked $Ub_3$ against disassembly by OTUB1 and inhibits but does not fully protect $Ub_4$ from the removal of a single Ub unit (Fig. 3c, d, also ref. 31). Combined with our IsoT cleavage results, this indicates that OTUB1 was able to cleave the unprotected distal Ub unit in the $Ub_4$:Ub4a complex, consistent with our NMR data.

We wished to investigate the effect of the proximal-Ub tail deletions in $Ub_4$ on the peptide's ability to interfere with polyUb disassembly by OTUB1. We found (Fig. 3e, f) that partial (ΔGG) deletion of the C-terminal tail in $Ub_3$ and $Ub_4$ did not alter how the peptide protects these chains from OTUB1 compared to their un-truncated counterparts (Fig. 3c, d). Indeed, the presence of Ub4a inhibited the removal of Ub from $[Ub]_3$-$Ub_{R74}$ and provided essentially full protection of the preassembled trimer, $[Ub]_2$-$Ub_{R74}$ (Fig. 3f), as well as the trimer resulting from the tetramer cleavage (Fig. 3e). On the other hand, deletion of the entire proximal Ub tail in the trimer, $[Ub]_2$-$Ub_{R72}$, resulted in complete loss of protection against disassembly by OTUB1 (Fig. 3g), consistent with the weakened Ub4a binding to $[Ub]_2$-$Ub_{R72}$ observed in our NMR studies (Fig. 1).

**Cyclic peptide Ub4a inhibits Ub chain elongation beyond trimers**

An essential initial step in Ub-mediated signaling is the formation of the polyUb signal itself. We anticipated that Ub4a binding to Ub chain can interfere with its recognition and elongation by E2/E3 enzymes, thus inhibiting chain growth above certain length. More specifically, based on our NMR data and DUB assays, we hypothesized that K48-linked Ub chain assembly reaction will not be affected by the peptide binding until the chain length reaches $n = 3$ after which further elongation would be impeded. To test this, Ub chain assembly assays were performed with and without the cyclic peptide.

As shown in Fig. 4, Ub chain assembly was not affected by the presence of Ub4a until the chain length reached $n = 3$, after which elongation to a tetramer was greatly retarded. By contrast, in the control reaction without the cyclic peptide (Fig. 4b) the chain assembly continued beyond the trimer stage. Furthermore, the dimer band seen in Fig. 4a persisted longer than in the absence of Ub4a (Fig. 4b). This suggests that Ub4a binds to $Ub_2$ to a certain degree until some amount of $Ub_3$ is formed, at which point the peptide binding shifts to tri-Ub. These results further corroborate our NMR and SPR[31] data that indicate Ub chain length selectivity of Ub4a, which binds $Ub_3$ and $Ub_4$

significantly tighter than $Ub_2$. These results also suggest that the inhibitory effect of Ub4a on the UPS can also be a result of the peptide's interference with the cell ubiquitination machinery causing shortage of polyUb chains that are longer than a trimer.

**Crystal structure of the $Ub_3$:Ub4a complex reveals the mechanism of interaction**

To gain detailed structural insights into the mechanism of interaction between Ub4a and K48-linked tri-Ub, we determined crystal structure of the $Ub_3$:Ub4a complex at 1.85 Å resolution. The crystals contained two molecules of $Ub_3$ and two molecules of Ub4a in the asymmetric unit. The structure is illustrated in Fig. 5 and S4, the data collection and structure refinement statistics are shown in Table S1. This structure revealed previously unavailable atomic-level details of the interactions between K48-linked tri-Ub and the cyclic peptide.

**General description of the $Ub_3$:Ub4a structure.** The three Ub units within each trimer form a ring-like arrangement and are mutually related by ~120° rotation around the axis passing through the ring's center (Fig. 5a). The hole at the center of the $Ub_3$ ring is occupied by Ub4a. Consistent with our previous NMR data[31], only one Ub4a molecule is bound to each tri-Ub. The two trimer complexes are very similar, with the r.m.s.d. between 1601 equivalent non-H atoms being 0.22 Å. The contacts between different Ub units and their interactions with Ub4a are common for both complexes (Tables S2, S3).

Despite the anisotropic mosaicity of the X-ray data, the final electron density maps were easily interpretable (see e.g., Fig. S4). Guided by the electron density, it was possible to model almost all residues in each $Ub_3$:Ub4a assembly. Only two C-terminal residues (G76 and D77) in the proximal Ub unit ($Ub_A$) were not modeled in one of the trimers. The C-terminus of this Ub interacts closely with the side chain of R72 of $Ub_C$ and with T12 of a symmetry-mate (this is a crystal-specific interaction) (Fig. S4c), leading to structural disorder. In fact, fragmented peaks of electron density suggest alternative traces for the C-terminal region of this Ub; however, they are too weak to complete the structural refinement. Importantly, all four covalent isopeptide linkages between Ub units in both $Ub_3$ chains are well defined by the electron density (Figs. 5c, d and S4b).

**Interactions between Ub units within $Ub_3$.** In addition to the $K48_{(N\zeta)} - {(CO)}G76$ isopeptide bonds that link subsequent Ub units within the tri-Ub chain, several noncovalent intra-chain interactions within the trimer define the relative positioning of adjacent Ub units. These interactions are detailed in Table S2 and illustrated in Fig. 5c-e. Despite the apparent 3-fold symmetry relating Ub units in the chain, their mutual interactions are not identical, even for the interfaces between adjacent Ub units, both formed around the isopeptide linkages through K48. Furthermore, the contacts between Ub units are quite limited (Fig. 5c, d, Table S2), and most extensive interactions are observed for the interface between $Ub_A$ and $Ub_C$ units that are not covalently linked. In this case, four H-bonds stabilize relative positions of these Ub units (Fig. 6e). It should be noted here that $D77_A$ is not a native residue to wild-type Ub and thus, the H-bond formed between this residue and $R72_C$ would not be present in the native chains. Our NMR data (Fig. S3d, e) along with DUB assays in which $D77_A$ was present (Fig. S12 in ref. 31) or absent (Fig. 3c, d) do not indicate a major difference in the interaction of Ub units with each other or with the peptide. Nevertheless, the presence of $D77_A$ in the C-terminal tail of the proximal Ub ($Ub_A$) unit does in fact result in an interesting interaction with $R72_C$ of the distal Ub ($Ub_C$) which might give the structure an additional anchor. It is also worth noting that the relative spatial arrangement of the three Ub units observed in this complex is almost certainly triggered by their interactions with the peptide, as discussed below.

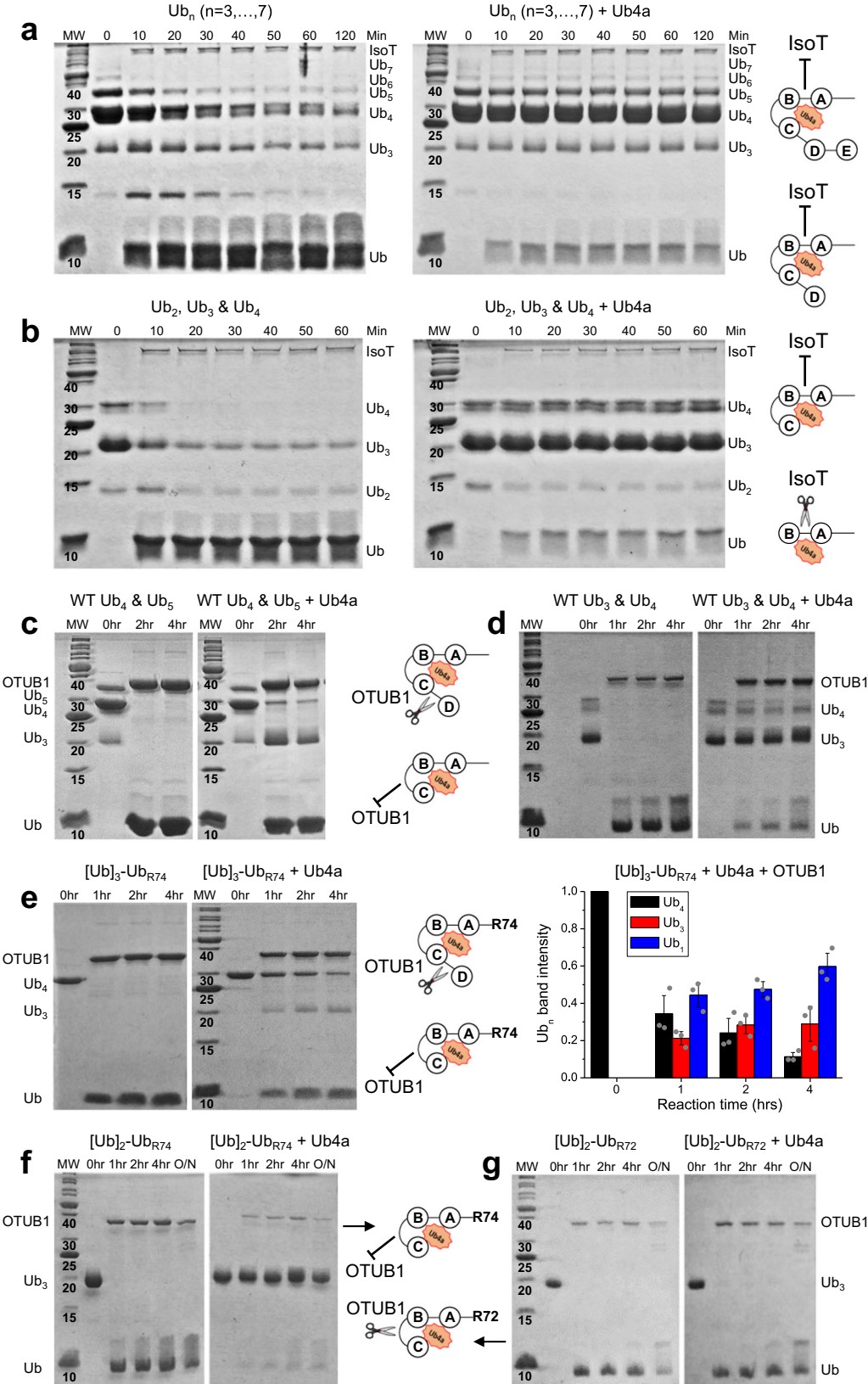

**Stabilizing interactions engaging Ub4a peptide.** These interactions can logically be partitioned into two categories. The first category includes stabilizing contacts within the Ub4a molecule itself. While they are naturally described here in the context of the complex, it is possible that some or many of them exist (stably or transiently) even in free, un-complexed, cyclic peptide. The cyclic peptide does not appear circular in shape but rather elongated and stabilized by five intramolecular H-bonds (magenta dashed lines in Fig. 5b). These include $Aoc5_{(O)}-_{(HN)}Thr8$, $Aoc5_{(NH)}-_{(O)}Thr8$, $Tyr3_{(O)}-_{(HN)}Val10$, $Tyr3_{(NH)}-_{(O)}Val10$, and $Gln2_{(OEl)}-_{(HN)}Gly12$ (linker). Interestingly, the topology of H-bonding inside the peptide is highly reminiscent of that found in an antiparallel β-sheet.

**Fig. 3 | Ub4a binding protects tri-Ub and longer chains against disassembly by DUBs.** SDS PAGE gels of the indicated time points of IsoT disassembly reactions for **a** mixture of K48-linked polyUb chains of length $n \geq 3$, and **b** mixture of $Ub_2$, $Ub_3$ and $Ub_4$ chains, alone (left) and in complex with Ub4a (right). SDS PAGE gels of select time points of OTUB1 disassembly reactions for K48-linked **c** $Ub_3$, $Ub_4$, and $Ub_5$; **d** $Ub_3$ and $Ub_4$; **e** $[Ub]_3$-$Ub_{R74}$, **f** $[Ub]_2$-$Ub_{R74}$, and **g** $[Ub]_2$-$Ub_{R72}$ alone and in complex with Ub4a. Molecular weight (MW) values are shown in kDa. The plot in **e** (right) shows normalized intensities of the $Ub_n$ gel bands at the indicated time points for $[Ub]_3$-$Ub_{R74}$:Ub4a disassembly by OTUB1 (see details in Methods). The

sum of normalized $Ub_1$, $Ub_3$, and $Ub_4$ band intensities equals one. The data for $Ub_1$, $Ub_3$, and $Ub_4$ are colored blue, red, and black, respectively. Shown are the mean values obtained in three attempts ($n = 3$) to quantitate gel bands using ImageJ; the error bars, centered on the mean values, represent standard deviations among these measurements. The gray dots depict the normalized intensities obtained in each of these attempts. The cartoon drawings depict the anticipated modes of peptide binding to and protection of the respective chains. In all the assays, Ub4a was added in equimolar ratio to polyUb. Source data are provided as a Source Data file.

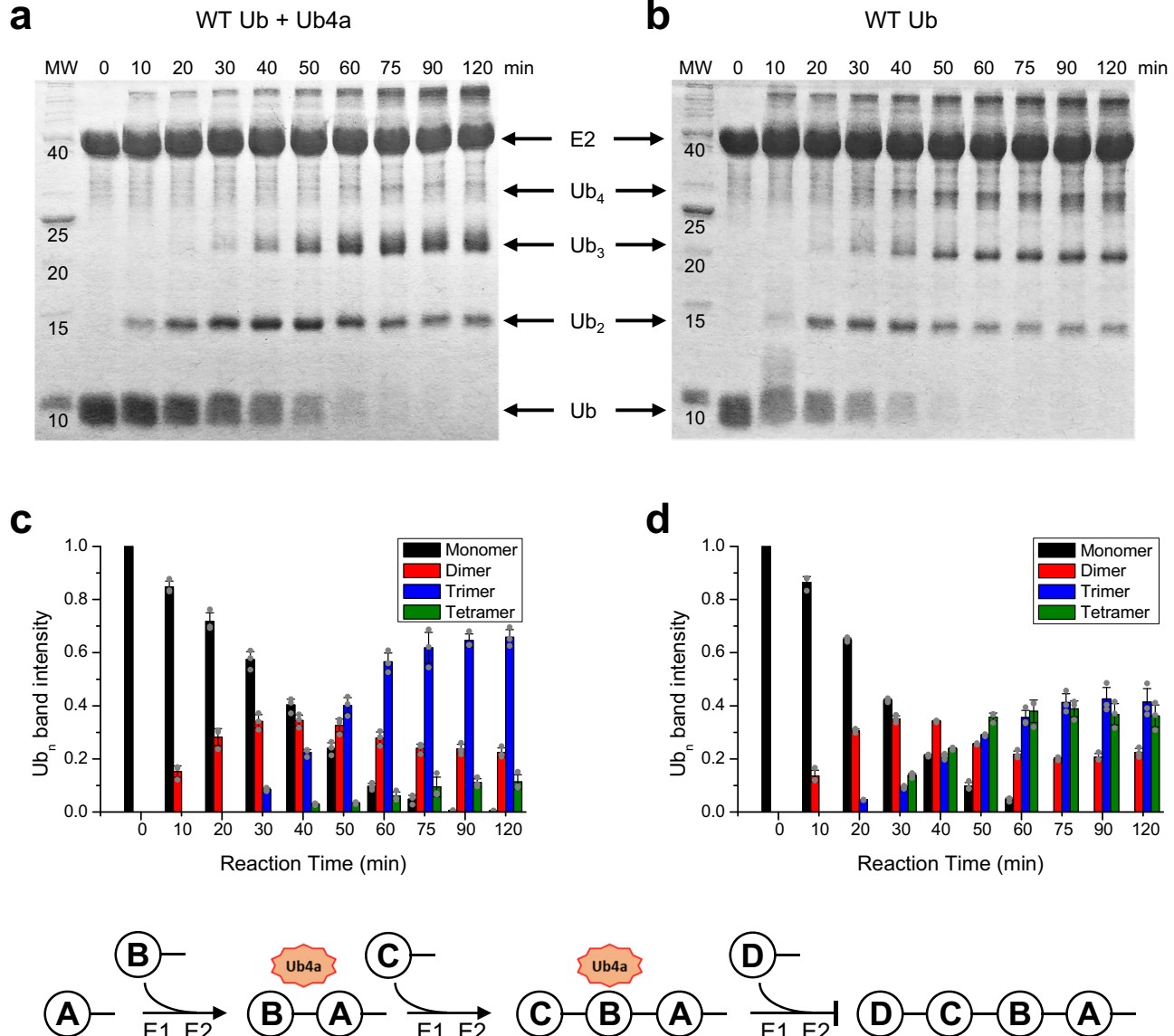

**Fig. 4 | The effect of Ub4a binding on the in vitro K48-linked enzymatic chain assembly.** SDS PAGE gels of K48-linked Ub chain assembly by E2-25K enzyme (GST-fused), and their quantitation. Shown are the results for select time points in the reaction (**a**) in the presence of equimolar (to Ub) amount of Ub4a and (**b**) without Ub4a, as control. Molecular weight (MW) values are shown in kDa. The normalized intensities of the $Ub_1$-$Ub_4$ bands are plotted in **c** and **d** respectively. The data for

$Ub_1$, $Ub_2$, $Ub_3$, and $Ub_4$ are colored black, red, blue, and green, respectively. Shown are the mean values obtained in three attempts ($n = 3$) to quantitate gel bands using ImageJ; the error bars, centered on the mean values, represent standard deviations among these measurements. The gray dots depict the normalized intensities obtained in each of these attempts. The cartoon on the bottom illustrates the chain assembly reaction. Source data are provided as a Source Data file.

The second category encompasses complementary intermolecular interactions between the peptide and tri-Ub which likely provide the major stabilization of the complex. Despite the small size of Ub4a, its interactions with $Ub_3$ are quite extensive (surface of the interface between the two entities, calculated with the PISA Server[41], is

~800 Å²), resulting from the fact that Ub4a occupies the central hole in $Ub_3$ and has comparable interactions with all three Ub units. The interface area can approximately be partitioned between three Ub subunits into 43% for $Ub_A$, 25% for $Ub_B$ and 32% for $Ub_C$. $Ub_3$ is tightly wrapped around the peptide, with all three Ubs oriented such that

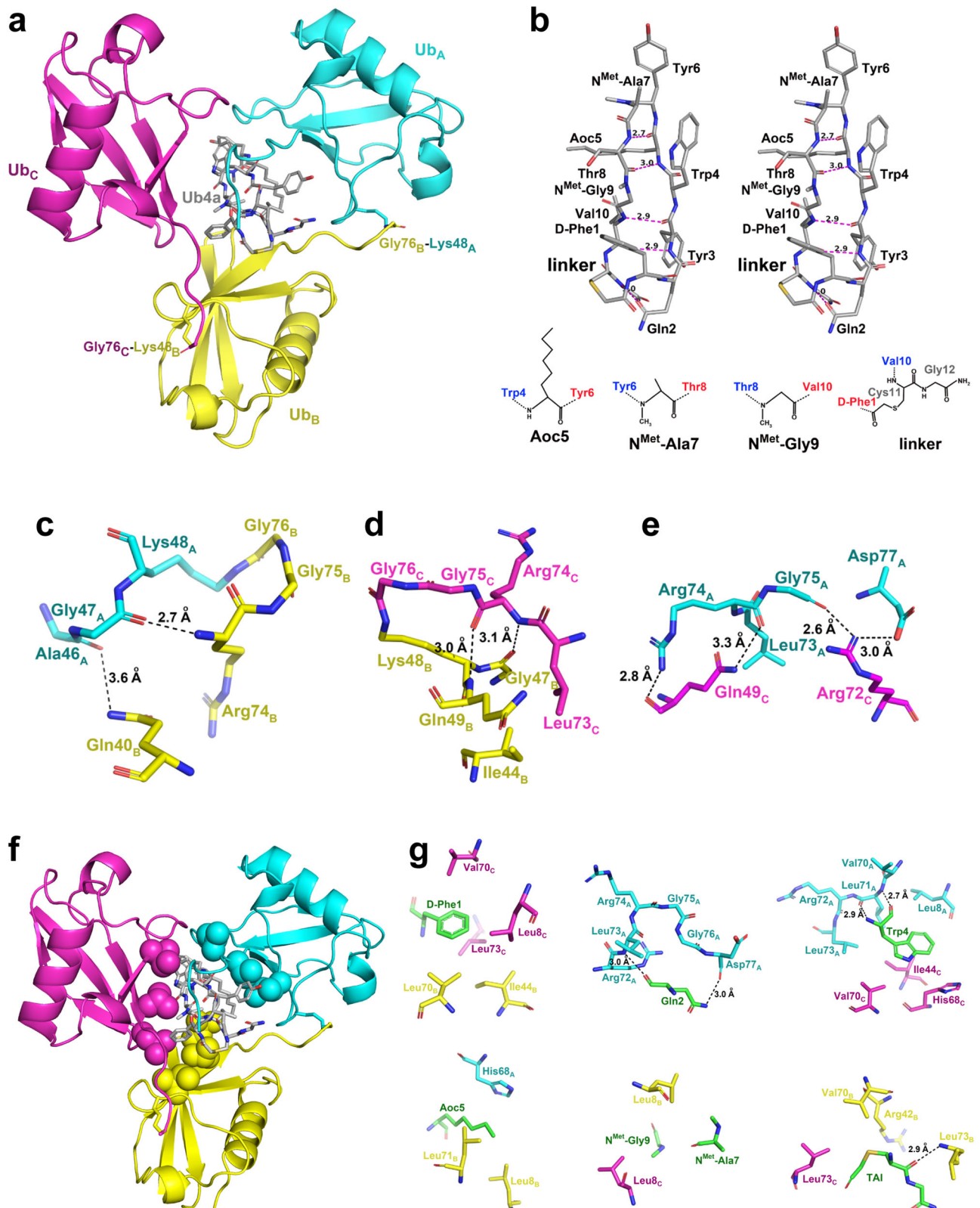

their hydrophobic patch residues are buried within the complex and facing the peptide (Fig. 5f). Specific Ub₃:Ub4a interactions are detailed in Table S3, and selected examples are illustrated in Fig. 5g. Even though the stabilizing interactions consist of a variety of H-bonds and hydrophobic contacts, it appears that the latter group is largely defining the strength of binding (affinity). Since most Ub interactions with ligands occur through Ub's canonical hydrophobic surface

patch[42], it is perhaps not surprising to find the peptide bound inside the tri-Ub ring and interacting with the hydrophobic patches of all three Ub units.

## Validation of the Ub₃:Ub4a crystal structure by solution NMR
To determine if the Ub₃:Ub4a structure observed in crystals is retained in solution we compared it with our NMR data. The residues in each Ub

**Fig. 5 | Crystal structure of the complex between K48-linked tri-Ub and Ub4a peptide.** In **a**, three covalently-linked units of Ub (drawn in a ribbon representation, colored cyan ($Ub_A$), yellow ($Ub_B$) and magenta ($Ub_C$) and annotated) are shown in complex with one molecule of Ub4a (drawn as gray sticks at the center). Ub residues (K48 and G76) forming covalent/isopeptide bonds that link subunits $Ub_A$ with $Ub_B$ and $Ub_B$ with $Ub_C$ are also shown in stick representation. Panel **b** shows a stereo view of the Ub4a molecule in the conformation found in the crystal structure of the complex with tri-Ub (see also Fig. S4). Five H-bonds formed by atoms within Ub4a are indicated by magenta dashed lines. Structural depictions of non-standard elements of Ub4a are illustrated under the stereo image. Interactions between Ub units within the $Ub_3$:Ub4a complex. Shown are interface contacts between (**c**) $Ub_A$ and $Ub_B$ or (**d**) $Ub_B$ and $Ub_C$ interface, as well as the interactions (**e**) between $Ub_A$

and $Ub_C$. Residues are shown in stick representation and labeled, and Ub units are distinctly colored ($Ub_A$ - cyan, $Ub_B$ - yellow and $Ub_C$ - magenta) as in **a**. H-bonds are indicated by black dashed lines, and the corresponding distances are shown. **f** The structure from **a**, with the hydrophobic patch residues L8, I44, V70 of each Ub shown in sphere representation; coloring as in **a**. **g** Representative examples of stabilizing intermolecular interactions between different elements of the Ub4a peptide and Ub units of tri-Ub. Residues are shown in stick representation and labeled. The fragments of Ub4a are colored green, while Ub residues are colored according to the respective Ub units, as in **a** and **f**. The peptide cyclization linker fragment (see **b**) is labeled TAI. H-bonds between Ub4a and $Ub_3$ are indicated by black dashed lines, and the associated distances are also shown. A detailed description of the intermolecular contacts is in Table S3.

unit that exhibited strong NMR signal perturbations (shifts and/or attenuations) upon Ub4a binding were mapped onto the structure (Fig. 6a). Most of the perturbed residues are located either at the $Ub_3$:Ub4a interface or at the interface between Ub units, thus validating the crystal structure. A few residues which are solvent exposed and relatively far from the binding interface also showed signal perturbations; these could reflect domain rearrangement/reorientation within tri-Ub caused by peptide binding in solution.

To further verify the $Ub_3$:Ub4a structure in solution we utilized a paramagnetic nitroxide spin label (TEMPO) covalently attached to a specific site on the peptide that is not in direct contact with $Ub_3$ ($Ub4a^{TEMPO}$, see Figs. 6 and S5 and Methods). The nitroxide induces distance-dependent paramagnetic relaxation enhancement (PRE) of nuclear spins located within ~25 Å from its unpaired electron[43]. This allowed us to determine/measure distances between TEMPO and individual residues in each Ub unit in the complex in solution.

The addition of $Ub4a^{TEMPO}$ to selectively [15]N-labeled $Ub_3$ resulted in strong residue-specific signal attenuations (PREs) in the proximal ($Ub_A$) and endo ($Ub_B$) Ub units in the trimer, with the least effect observed for the distal Ub ($Ub_C$) (Figs. 6b, S6). These effects were particularly pronounced for the C-termini of all three Ub units and residues around K48 of the proximal and endo Ubs, which are involved in isopeptide linkages. To quantify our observations, we used experimental PREs and atom coordinates from the $Ub_3$:Ub4a crystal structure to reconstruct the position of the unpaired electron. The location of TEMPO's unpaired electron obtained from a global fit of the PREs observed in all three Ubs together was within ~7 Å from its expected position based on the modeled structure of $Ub4a^{TEMPO}$ (Fig. 6c), and within 4–5 Å from the locations derived using PREs in the proximal or endo Ubs analyzed separately. This is a remarkable agreement, given the uncertainty in the electron's position due to the intrinsic flexibility of the paramagnetic tag. It is worth noticing that the $G76_C$–$K48_B$ and $G76_B$–$K48_A$ isopeptide linkages and the C-terminus of $Ub_A$ are all located on the side of the tri-Ub ring facing TEMPO, which explains the abovementioned strong PREs in these groups. Together, these results further corroborate the conclusion that the crystal structure faithfully represents the $Ub_3$:Ub4a complex in solution.

### Insights into Ub4a binding to substrate-conjugated ubiquitin chains

The studies of Ub trimers and tetramers described above were performed using free Ub chains in which the C terminus of the proximal Ub is unanchored. In cells, Ub chains are often tethered to other proteins through an isopeptide bond involving the C-terminal G76 of the proximal Ub, raising the question if conjugation of polyUb to a substrate can affect the ability of the peptide to bind to that chain. To examine whether substrate anchoring of polyUb can obstruct cyclic peptide's access to (and interaction with) the C-terminal tail of the proximal Ub in the tethered chain, we took advantage of our NMR and DUB data (Figs. 1, 3) that suggest that full truncation of the C-terminal tail renders Ub unrecognizable as "true" Ub by Ub4a. Thus we used $[Ub]_3$-$Ub_{R72}$ as a mimic of tri-Ub tethered to a model substrate (in this

case $Ub_{R72}$). We made two $[Ub]_3$-$Ub_{R72}$ chains, one with [15]N-labeled proximal Ub ($[Ub]_3$-[15]N$Ub_{R72}$) and the other with [15]N-labeled next-to-proximal, $Ub_B$ unit ($[Ub]_2$-[15]N$Ub_B$-$Ub_{R72}$) and used NMR to examine peptide's interactions with these constructs. The addition of Ub4a caused dramatic changes in the NMR spectra of $[Ub]_2$-[15]N$Ub_B$-$Ub_{R72}$ (Fig. 7a), including large signal shifts in the hydrophobic patch residues and large shifts (L73) or signal disappearance (R72, R74, G75, G76) in the C-terminal tail of that Ub unit (Fig. 7c). Remarkably, the spectrum of that $Ub_B$ unit in the Ub4a bound state was essentially the same as of the Ub4a-bound proximal Ub in the un-truncated tri-Ub (Figs. 7e and S7). By contrast, only minor NMR signal perturbations were detected in the proximal $Ub_{R72}$ in $[Ub]_3$-$Ub_{R72}$, as anticipated from and in full agreement with the abovementioned observations for the distal Ub in that chain. The only significant perturbations were a strong signal shift of the side chain NH group of K48 which is directly involved in the isopeptide bond, and noticeable (~9 and 7 fold) signal attenuations in the adjacent to it residues A46 and G47. Combined with the results of our NMR studies of the distal Ub and DUB assays, these data provide direct evidence of the shift in the peptide-binding mode/site upon deletion of the entire C-terminal tail of tetra-Ub. Furthermore, these observations corroborate our assumption that $Ub_{R72}$ is not fully recognized by Ub4a as 'ubiquitin', and therefore acts as a model substrate in that chain. Importantly, the NMR signal perturbations detected both in the C-terminal tail of $Ub_B$ and in the side chain of K48 of $Ub_{R72}$ indicate that the peptide has access to and interacts directly with the linkage between polyUb (in this case $Ub_3$) and the substrate (in this case, $Ub_{R72}$). This suggests that the peptide-binding mechanism revealed in our study has direct relevance to peptide's interaction with a substrate-tethered polyUb chain.

## Discussion

Post-translational modification of proteins by Ub or Ub chains is an essential cellular regulatory mechanism involved in a host of biological processes, including signal transduction, transcriptional regulation, cell cycle, growth control, apoptosis and oncogenesis. Consequently, Ub and polyUb are promising robust targets for therapeutic and ameliorative intervention. Our previous efforts had resulted in discovery of proteinogenic and non-proteinogenic macrocyclic peptides as agents that selectively and tightly bind and modulate long ($n \geq 3$) K48-linked Ub chains that act as the primary signal for proteasomal protein degradation[11,31]. More recently, Vamisetti et al. also discovered macrocyclic peptides with high affinities and selectively for K63-linked di-Ub which were found to modulate DNA repair processes in cells[44]. Our studies described herein uncovered the structural mechanism by which the non-proteinogenic cyclic peptide Ub4a selects a particular Ub trimer moiety in a K48-linked tetra-Ub chain and the extent to which the C-terminal tail of the chain determines the selection of the binding site and affects the overall strength of peptide binding and ultimately the chain protection from disassembly by DUBs.

Our NMR data combined with Ub chain assembly and disassembly assays indicated that Ub4a selectively binds to the proximal trimer moiety within tetra-Ub. The crystal structure of $Ub_3$:Ub4a complex,

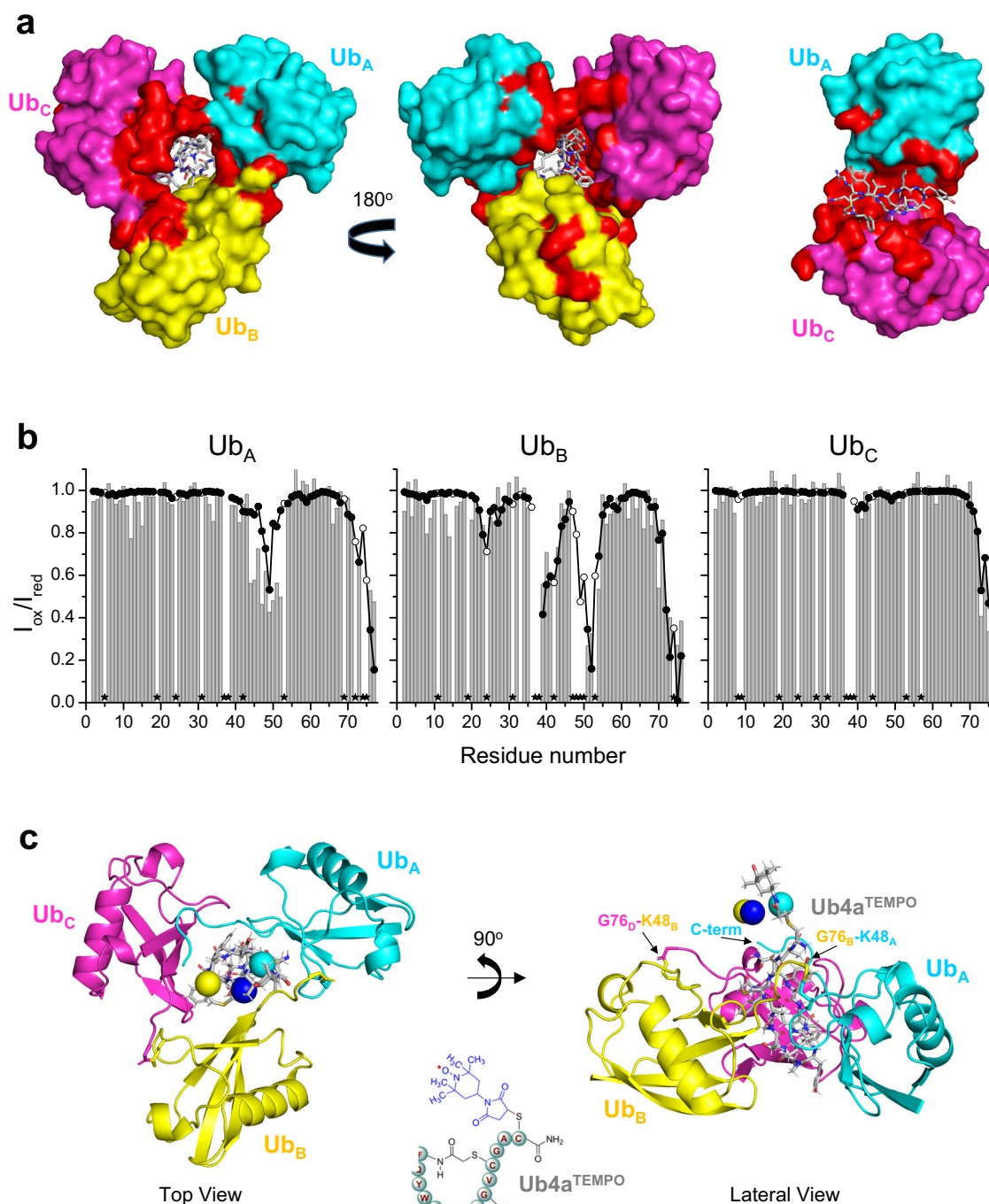

**Fig. 6 | Solution NMR-based validation of the crystal structure of the Ub3:Ub4a complex. a** Residues in each Ub unit that exhibited strong NMR signal perturbations (CSP > mean + 0.5 SD or complete signal attenuation) mapped (colored red) on the crystal structure of the Ub3:Ub4a complex, where Ub3 is shown as a surface, front and back view. The Ub units are colored as in Fig. 5: $Ub_A$ - cyan, $Ub_B$ - yellow and $Ub_C$ – magenta, the Ub4a peptide is shown as gray sticks. Shown on the right is a 'slice' view of the inside of the central hole in the complex with the $Ub_B$ unit artificially removed. **b** PRE effects (signal attenuation, $I_{ox}/I_{red}$, represented by bars) in the individual Ub units of Ub3 caused by $Ub4a^{TEMPO}$ (Fig. S5) and the results of their (global) fit (shown as solid circles) to determine a single position of the paramagnetic moiety in 3D space (see also Fig. S6). Open circles show predictions

for those residues/amides (marked with stars) where experimental data were not available. **c** Reconstructed position (shown as sphere) of the TEMPO's unpaired electron (from the fits in **b**) superimposed on the crystal structure of Ub3:Ub4a complex in which Ub4a was replaced with $Ub4a^{TEMPO}$. Spheres correspond to the results of fitting PRE data for all three Ub units together (global fit, blue sphere) and also for the proximal Ub (cyan sphere) and middle (endo) Ub (yellow sphere) separately. For illustrative and comparison purposes $Ub4a^{TEMPO}$ was modeled in PyMOL[51] by building TEMPO on Ub4a from the Ub3:Ub4a complex structure. The individual Ub units are colored as in **a**: $Ub_A$ - cyan, $Ub_B$ - yellow and $Ub_C$ – magenta; $Ub4a^{TEMPO}$ is shown as gray sticks. The chemical structure of $Ub4a^{TEMPO}$ is drawn on the bottom. Source data are provided as a Source Data file.

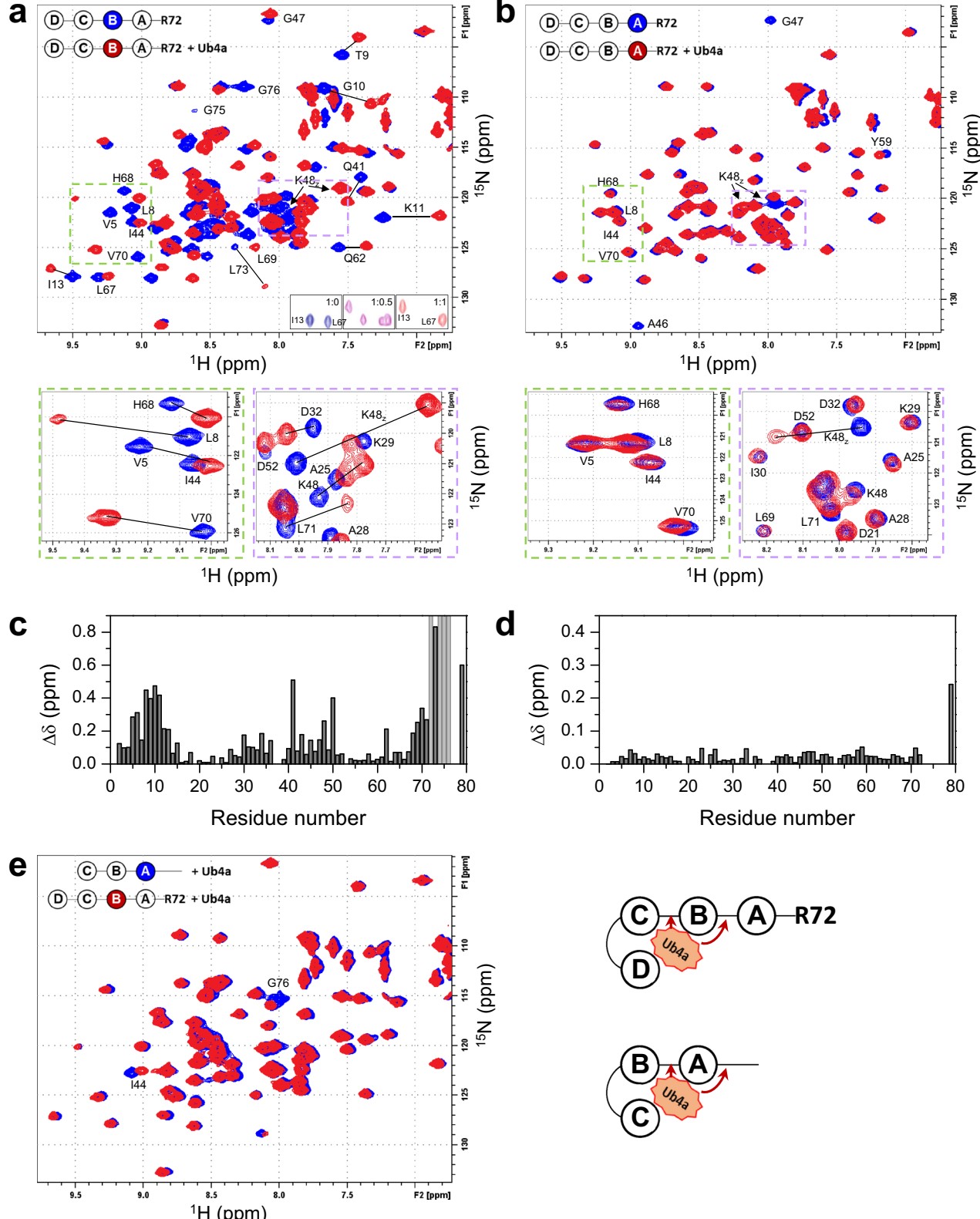

validated by solution NMR measurements, revealed how this binding occurs. Specifically, all three Ub units wrap around the peptide in a ring-like arrangement enabling their surface hydrophobic patch residues, positioned on the inner surface of the ring, to contact the peptide placed inside the ring (Fig. 5). This spatial arrangement enables the peptide to engage all three Ub units. In addition to the hydrophobic contacts at the interface, the complex is further stabilized by residue-

specific polar interactions between the peptide and Ub units (Fig. 5g) and between Ub units themselves (Fig. 5c-e). In particular, H-bonds between $Ub_A$ and $Ub_C$ ($G75_A$−$R72_C$ and $R74_A$−$Q49_C$) and those between the C terminus of $Ub_A$ and Ub4a ($L71_A$−Trp4, $L73_A$−Gln2, as well as $D77_A$−Gln2) appear to lock the $Ub_3$ ring with the cyclic peptide being held inside it. These interactions highlight the involvement of the tail of the proximal Ub in the peptide binding and explain the

**Fig. 7 | Ub4a binding to the endo (Ub$_B$) and proximal Ub units of tetra-Ub chain with entire C-terminal tail deletion (ΔLRGG). a** Overlay of $^1$H-$^{15}$N SOFAST-HMQC NMR spectra of the endo/next-to-proximal Ub (Ub$_B$) in [Ub]$_2$-$^{15N}$Ub$_B$-Ub$_{R72}$ free in solution (blue) and upon addition of Ub4a (red) in 1.5:1 molar ratio (peptide:polyUb). **b** Overlay of $^1$H-$^{15}$N SOFAST-HMQC NMR spectra of the proximal Ub (Ub$_{R72}$) in [Ub]$_3$-$^{15N}$Ub$_{R72}$ free in solution (blue) and upon addition of Ub4a (red) in 2:1 molar ratio (peptide:polyUb). Spectral regions containing signals of the hydrophobic patch residues (L8, I44, V70) and the K48 isopeptide-bond signal (K48$_Z$) are marked using orange and violet rectangles, respectively. Zoom in on these regions are shown below the respective full spectra. Signals of select residues are indicated, and their shifts are highlighted using straight lines. Insets on the bottom of the spectrum in **a** depict NMR signals of I13 and L67 in Ub$_B$ at [Ub]$_2$-$^{15N}$Ub$_B$-Ub$_{R72}$:Ub4a molar ratios of 1:0 (blue), 1:0.5 (magenta), and 1:1 (red), to illustrate the slow-exchange regime of binding. **c** Residue-specific chemical shift perturbations (CSPs, $\Delta\delta$) in the Ub$_B$ unit of [Ub]$_2$-$^{15N}$Ub$_B$-Ub$_{R72}$ at the endpoint of titration with Ub4a (from **a**). Signals of C-terminal residues marked with light gray bars disappeared entirely. **d** Residue-specific CSPs in the proximal Ub (Ub$_{R72}$) of [Ub]$_3$-$^{15N}$Ub$_{R72}$ at the endpoint of titration with Ub4a (from **b**). The CSP of the K48 side-chain NH group involved in the isopeptide bond is shown at residue position 79. Note the different vertical scales of the plots in **c** and **d**. **e** Overlay of $^1$H-$^{15}$N SOFAST-HMQC NMR spectra of the proximal Ub (Ub$_A$) in tri-Ub (blue) and of Ub$_B$ in [Ub]$_2$-$^{15N}$Ub$_B$-Ub$_{R72}$ (red) at the endpoint of titration with Ub4a. Note the similarities between the blue and red spectra. The cartoon drawings to the right of **e** illustrate the peptide binding arrangements deduced from these NMR studies and highlight the interactions of Ub4a with the C-terminal tails and the isopeptide bonds of the indicated Ub units revealed by these data. Source data are provided as a Source Data file.

selection of the proximal Ub trimer moiety by the peptide. These conclusions agree with our proximal-tail truncation data. Indeed, the deletion of residues L73$_A$-G76$_A$ (ΔLRGG), eliminating the above-mentioned H-bonds holding the tri-Ub ring wrapped around the peptide, significantly weakened binding to tri-Ub and resulted in a shift of the peptide-binding site (trimer moiety) toward the distal end of the tetra-Ub chain (Fig. 1). By contrast, the removal of G75$_A$ and G76$_A$ (ΔGG) had only a minor effect on the binding (Fig. 2).

The studies described here used free Ub chains in which the C terminus of the proximal Ub is unanchored. While free/unanchored polyUbs, including K48-linked chains, have been recently reported to play important physiological roles in cells[45,46], most of known Ub-mediated signaling involves Ub chains conjugated to other proteins through the C-terminal G76 of the proximal Ub. The tethering could potentially sterically obstruct the peptide's access to and interactions with the C-terminal tail of the polyUb tag. However, our NMR data for the [Ub]$_3$-Ub$_{R72}$ construct that mimics tri-Ub conjugated to a model substrate (Ub$_{R72}$) demonstrate that Ub4a has access to both sides of the Ub$_3$–substrate linkage, namely the C-terminal tail of the next-to-proximal Ub unit (Ub$_B$) and the K48 side-chain of Ub$_{R72}$ involved in the isopeptide bond. Furthermore, the NMR spectra of Ub$_B$ in the Ub4a-bound state are essentially the same as of the peptide-bound proximal Ub in unanchored tri-Ub, suggesting similar interactions with the peptide. Notably, our structure of the Ub$_3$:Ub4a complex shows no contacts between the peptide and G76$_A$, while a polar interaction with D77$_A$ is present only in one of the two complexes in the asymmetric unit. The fact that these two C-terminal residues are disordered in one of the complexes also indicates that they are not major contributors to the peptide binding. This conclusion is further supported by our NMR and DUB data for Ub chains with partially and fully deleted C-terminal tail that point to residues L73 and R74 playing role in the peptide's selection of the proximal trimer. It is also worth emphasizing here that even if tethering to a substrate renders the entire C-terminal tail of the proximal Ub inaccessible, this does not abrogate Ub4a binding to long polyUb chains. In fact, our NMR and DUB assays for tetra-Ub with a fully deleted C-terminal tail demonstrate that for polyUb chains of length $n > 3$, required for efficient proteasomal targeting[27], obstructing peptide's access to the C-terminal tail of the proximal Ub will merely cause a shift of the binding mode/site towards the distal end of the chain with little effect on the strength of binding or the peptide's ability to prevent polyUb chain recognition by the proteasomal receptors or cleavage/disassembly by DUBs. Overall, these observations support the relevance of our results to substrate-conjugated polyUb chains.

It has previously been shown that treatment of mice with Ub4a inhibits growth of tumors in vivo[31] indicating the therapeutic effect of Ub4a in cancer treatment through interference with the UPS. Here, our in vitro chain elongation assay has shown that in the presence of Ub4a enzymatic extension of K48-linked Ub chain is hampered after reaching a trimer length. This suggests that, in addition to direct protection from DUBs or proteasomal receptors, a possible way by which Ub4a can affect proteasomal degradation of oncogenic proteins is through blocking K48-linked tri-Ub chains from elongation, thereby reducing the availability of tetra-Ub and longer chains required for efficient degradative signaling. In cases where tetra-Ub and longer chains are synthesized, tight Ub4a binding to the proximal trimer moiety of the chain can block it from recognition by receptors as well as protect from cleavage by DUBs; the latter would cause a decrease in the pool of available free Ub monomers that can be used for other cellular signaling purposes. Thus, we speculate that Ub4a binding to K48-linked Ub chains can trigger a cumulative inhibition effect on Ub-mediated signaling, thus making it a promising avenue for cancer therapy.

Structural data obtained in this study highlight the conformational malleability or K48-linked polyUb chains that in the unbound state adopt predominantly compact conformations driven by hydrophobic Ub-Ub contacts (as seen for di-Ub[47,48] and tetra-Ub[48,49], also Fig. S8) but open and rearrange to accommodate Ub4a (as illustrated in Fig. S9) or other binding partners (e.g., ref. 50). The structural details of the Ub$_3$:Ub4a complex provide mechanistic insights into the underlying interactions and how they can be modulated by side-chain modifications in the peptide. For example, another macrocyclic peptide, Ub4e, that differs from Ub4a in 5 out of 12 residues (Fig. S10) has similarly strong selectivity for K48-linked tetra-Ub albeit a somewhat lower affinity (twofold higher $K_d$) compared to Ub4a[31]. Based on our Ub$_3$:Ub4a structure we modeled the Ub$_3$:Ub4e complex using in-silico mutagenesis in PyMol[51]. This structure model (Fig. S10b) shows that almost all key contacts (from Table S3) between Ub$_3$ and the peptide are preserved, and the side chains that are different between the two peptides are either solvent exposed (Tyr6/His6) or are capable of forming hydrophobic contacts (Thr8/Ile8, Val10/Leu10) with the same Ub units. However, the replacement of Tyr3 with Leu3 results in a loss of the H-bonds involving the hydroxyl group of the Tyr3 (Fig. S10b), which might explain the weaker binding of Ub4e. Thus, we expect that the structural and biochemical data obtained here will enable structure-based design of new peptide analogues with improved binding and physiochemical properties and for basic research.

## Methods

### Materials

All Ub monomers were expressed in *E. coli* BL21(DE3) strain carrying a helper plasmid pJY2[52] and purified as described previously[43,48,50]. The amino acid sequences of Ub variants used here are shown in Fig. S3.

### PolyUb chains assembly

PolyUb chains were made using controlled-length enzymatic assembly that allowed isotopic enrichment of selected Ub units[43,48,53]. The K48-linkage-specific conjugating E2 enzyme E2-25K (Ube2K)[48] was used together with Ub activating E1 enzyme (Ube1) to form K48-linked Ub chains. Specific Ub mutants, Ub K48R (distal) and Ub D77, ΔGG, or ΔLRGG (proximal), were used to control the resulting Ub conjugates

and to ensure proper incorporation of $^{15}$N-labeled Ub units at each step by preventing uncontrolled chain elongation[48,53]. Ub chain assembly reactions were carried out overnight at 37 °C using equimolar amounts of respective Ub species (1 mM), catalytic amounts of E1 (100 nM) and E2-25K (20 μM), and 2 mM ATP in a 2 mL, pH 8.0 reaction mixture[53]. Reaction products were separated by cation-exchange chromatography using 50 mM ammonium acetate, pH 4.5 as the equilibration buffer and 50 mM ammonium acetate, 1 M sodium chloride, pH 4.5 as the elution buffer. Electrospray ionization mass spectrometry (ESI-MS) was used to verify each product. The D77 extension was removed, when necessary, by incubating polyUb samples with yeast ubiquitin hydrolase-1 (YUH1)[53] overnight at 37 °C, and the removal of D77 was confirmed by mass spectrometry. YUH1 was removed by cation-exchange chromatography.

## Synthesis of macrocyclic peptides

Solid-phase peptide synthesis (SPPS) was employed for the synthesis of peptides manually in teflon filter equipped syringes, purchased from Torviq, or by using an automated peptide synthesizer (CS336X, CSBIO). Unless specified, all the chemicals used were analytical grade. N, N-dimethylformamide (DMF), Diisopropylethylamine (DIEA), Dichloromethane (DCM), and Trifluoroacetic acid (TFA), were purchased from Biolab. 2-Chloroacetic acid was purchased from Acros Organics. Diisopropyl azodicarboxylate (DIAD), 2-Mercaptoethanol, Methanol, TEMPO (4-maleimido-2,2,6,6-tetramethyl-1-piperidinyl oxy) free radical, N-Methyl-2-Pyrrolidone, 2-Nitrobenzenesulfonyl chloride and Palladium (II) chloride (PdCl2) were purchased from Sigma-Aldrich. Resins were purchased from CreoSalus. All coupling reagents 1-[bis(dimethylamino)methylene]−1H-1,2,3-triazolo[4,5-b]pyridinium 3-oxid hexafluorophosphate (HATU), [(6-chlorobenzotriazol-1-yl)oxy-(dimethylamino)methylidene]-dimethylazanium hexafluorophosphate (HCTU), and Hydroxybenzotriazole (HOBt) were purchased from Luxembourg Bio Technologies and GL Biochem. 9-fluorenylmethoxycarbonyl (Fmoc) protected amino acids were purchased from GL Biochem. Triisopropylsilane (TIPS) and Dithiothreitol (DTT) were purchased from Alfa Aesar. Fmoc-2-AoC was purchased from Advanced ChemTech. Triphenylphosphine(PPh3) and 2,4,6-Collidine were purchased from Tzamal D-Chem. Analytical high-performance liquid chromatography (HPLC) was performed on a Thermo instrument (Dionex Ultimate 3000) using Xbridge (4.6 × 150 mm, 3.5 μm, BEH300 C4, waters) column and analytical XSelect (waters, CSH C18, 3.5 μm, 4.6 × 150 mm) column with flow rate of 1.2 ml/min. Thermo Scientific instrument (Dionex Ultimate 3000) used Jupiter C18 (Phenomenex, 10 μm, 300 Å, 250 × 10 mm) column and Jupiter C4 (Phenomenex, 10 μm, 300 Å, 250 × 10 mm) column with a flow rate of 4 mL/min. Preparative HPLC was performed on a Thermo Scientific instrument (Dionex Ultimate 3000) using XSelect (waters, CSH C18, 10 μm, 19 × 250 mm) with a flow rate of 15 mL/min. HPCL was used for the purification of all peptides (HPLC mobile phases: buffer A: 0.1% TFA in H$_2$O and buffer B: 0.1% TFA in CH$_3$CN) and were characterized by ESI-MS using Thermo Scientific LCQ Fleet mass spectrometer with an ESI source. Further details can be found in Supplemental Information (Fig. S5).

## Crystallization, data collection and structural refinement

Crystals of the Ub$_3$:Ub4a complex were grown from the hanging droplets composed of equal volumes of Ub$_3$:Ub4a (8 mg/ml) and the crystallization solution containing 0.15 M NaCl, 23% (w/v) PEG 3350 and 0.1 M HEPES (pH 7.5). Incubated at 20 °C, crystals appeared within 3-5 days and continued to grow for additional 20-30 days. For X-ray data collection, crystals were transiently transferred to the solution containing the abovementioned components with part of water replaced by glycerol (to the final concentration of 25% v/v) and Ub4a (to a final concentration of 5 μM). Subsequently, crystals were frozen in liquid nitrogen, followed by collection of X-ray data.

The X-ray data were collected at the Advanced Photon Source, Argonne National Laboratory (Argonne, IL, USA), on beamline 22-ID, using 1.000 Å wavelength, at 100 K temperature. All tested crystals displayed significant anisotropic mosaicity, a problem that was circumvented by acquisition of narrow-oscillation (0.15˚) individual images. Images were processed and scaled with HKL-3000 suite v720[54].

Structure of the complex was solved by molecular replacement (MR) with the program Phaser[55] using the monomer A extracted from the structure deposited in the Protein Data Bank (PDB ID 1ubq)[56]. The MR showed unambiguous solution for six Ub units, arranged as two trimers. The MR solution was first subjected to rigid-body refinement at the resolution of 2.5 Å with the program REFMAC5[57], followed by several cycles of a structural refinement of positions and isotropic atomic displacement parameters (B's) for non-H atoms. At this stage, traces of covalent linkages between the Ub units within each of the two trimers became quite apparent. Also, the difference electron density peaks (F$_o$-F$_c$) clearly indicated the presence of unaccounted molecular content. Guided by this electron density, it was possible to model one molecule of Ub4a within each Ub$_3$ complex (Fig. S4c). The restraint libraries for non-standard linkages between Ub units, as well as unnatural fragments of Ub4a, needed for subsequent structural refinement, were created with the aid of the program JLigand[58]. The refinement was carried out to the extent of experimental data (i.e., the resolution of 1.85 Å). Emerging model was intermittently inspected with the aid of the program Coot[59] v0.9.6 and appropriate corrections were introduced. At the final stages of the refinement, we could identify and model two molecules of the HEPES buffer (one per trimer) and one molecule of glycerol (both were components of the crystallization or cryo-protectant solution). Additionally, 302 water molecules could be modeled within the asymmetric unit, based on the difference electron density peaks. Final models were evaluated by the MolProbity server[60] and validated with the Protein Data Bank validation pipeline. Images were created using PyMol[51] v.2.5.2.

Diffraction intensities were analyzed for the presence of twinning, using the procedure Truncate[61] within the CCP4[62] suite v7.1, and the diffraction anisotropy, using the Web-server https://services.mbi.ucla.edu/anisoscale/. Neither of these effects was identified. Due to the significant mosaicity of the diffraction pattern, mentioned earlier, experimental intensities could be integrated and scaled with only modest statistics (see Table S1). Furthermore, the translational pseudo-symmetry (tNCS, 0.5, x, 0.5) was detected in the final data. This symmetry closely coincides with the Patterson symmetry of the P2 space group, and it is well established that such a correlation can greatly hinder structural refinement (due to an ambiguity between crystallographic and NCS symmetries). Therefore, the statistics associated with the final refined model may be worse than expected based on the resolution and quality of the experimental data[63], as evident from Table S1. Ramachandran statistics for the Ub$_3$:Ub4a structures: 461 (99.6%) residues with backbone (ϕ, ψ) torsion angles in the preferred regions, 2 (0.4%) in the allowed regions, and 1 (0.2%) outlier.

We also attempted to co-crystallize Ub$_4$ with a cyclic peptide. However, the peptide was excluded from the crystals during crystallization, leaving us with the crystals of pure Ub$_4$. These crystals diffracted at 1.7 Å resolution (Table S1) and revealed a compact structure of Ub$_4$ (PDB ID 8e7o) that is generally similar (Cα r.m.s.d. of 0.837 Å) to the previously published Ub$_4$ structure[49] (PDB ID 2o6v), see Fig. S8. The X-ray data were collected at the Advanced Photon Source, Argonne National Laboratory (Argonne, IL, USA), on beamline 24-ID-E, using 0.97918 Å wavelength and 0.3˚ oscillation per individual image, at 100 K temperature. Ub$_4$ structure was solved by MR using the structure of K48-linked Ub$_2$ (PDB ID 1aar). Ramachandran statistics for the Ub$_4$ structure: 293 (99%) residues with backbone (ϕ, ψ) torsion angles in the preferred regions and 3 (1%) in the allowed regions. Images were created using PyMol[51] v.2.5.2.

## Ub chain disassembly and formation assays

Reactions of Ub chain disassembly by DUBs were performed as described[64] by adding OTUB1 or IsoT/USP5 (3 µM) to polyUb chains (35 µM) in 50 mM Tris buffer (pH 8.0) at 30 ˚C. Reaction samples were taken at various time points and quenched with SDS-PAGE loading dye, then resolved on a 15% polyacrylamide gel by SDS-PAGE, visualized by Coomassie blue staining and quantified using ImageJ 1.53[65].

To determine the effect of Ub4a binding on polyUb chain formation, monomeric Ub was incubated with equimolar (35 µM) or 2x (70 µM) amounts of Ub4a followed by a chain assembly reaction catalyzed by E1 (2 nM) and E2-25K (0.4 µM) in a 60 mL, pH 8.0 mixture. Samples were taken at select time points and analyzed using SDS PAGE gel visualization by Coomassie blue staining followed by quantification of the resulting bands using ImageJ 1.53[65]. In parallel, a similar assay was performed for Ub without adding Ub4a, as control.

## NMR experiments

NMR studies were performed at 23 °C on Bruker Avance III 600 MHz and 800 MHz spectrometers equipped with TCI and QCI cryoprobes, respectively. NMR samples containing Ub chains were prepared in 20 mM sodium phosphate buffer (pH 6.8) containing 5–10% $D_2O$ and 0.02% (w/v) $NaN_3$. Concentrated peptide stock solution was prepared by dissolving Ub4a or Ub4a$^{TEMPO}$ in 40% $d_6$-DMSO, 30% $D_2O$ and 30% 20 mM sodium phosphate buffer (pH 6.8). NMR experiments used standard Bruker pulse sequences (TopSpin v3.6.5). NMR data were processed using TopSpin v4.1.4 (Bruker Biospin) and analyzed with NMRFAM-Sparky 1.47 powered by Sparky 3.190[66]. Typically, the NMR spectra were acquired with spectral widths of 9615 Hz for $^1$H and 2128 Hz for $^{15}$N (at 600 MHz) and 12820 Hz for $^1$H and 2838 Hz for $^{15}$N (at 800 MHz); the acquisition time of ca. 80 ms, and 32 to 256 scans. NMR signal assignment of polyUb in the Ub4a-bound state was performed using BMRB Entry IDs 52127 and 52139 as the starting point, and for Ub signals in slow exchange utilized a combination of 'minimum signal shift' concept and $^{15}$N-edited TOCSY (mixing time 19-65 ms) and NOESY (mixing time 200-220 ms) spectra recorded at the titration endpoint. Identification of Ub side-chain signals was guided by BMRB entry ID 17769.

Binding studies were conducted by titrating increasing amounts of Ub4a into solution of polyUb with specific Ub unit $^{15}$N enriched, and binding was monitored by recording $^1$H-$^{15}$N SOFAST-HMQC spectra at each titration point. The starting concentration of polyUb was at 200 µM, except for [Ub]$_3$-$^{15N}$Ub$_{R72}$ and [Ub]$_2$-$^{15N}$Ub$_B$-Ub$_{R72}$ studies where the starting tetra-Ub concentration was 100 µM and [Ub]$_2$-$^{15N}$Ub where is was 150 µM. Stock concentrations of Ub4a ranged from 1.5 mM to 5 mM. We previously verified that the presence of DMSO in the amounts similar to those added with the peptide has negligible effect on $^{15}$N-Ub spectra[11].

Amide NMR signal shifts were quantified as chemical shift perturbations (CSPs), calculated for each residue as: $\Delta\delta = [(\delta_{HB}-\delta_{HA})^2 + ((\delta_{NB}-\delta_{NA})/5)^2]^{1/2}$, where $\delta_H$ and $\delta_N$ are chemical shifts of $^1$H and $^{15}$N resonances, respectively, for a given NH group, and A and B refer to the unbound and bound species, respectively.

For paramagnetic relaxation enhancement (PRE) measurements, $^1$H-$^{15}$N HSQC spectra of selectively $^{15}$N-labeled polyUb (100 µM) in a 1:2 molar mixture with Ub4a$^{TEMPO}$ (200 µM) were recorded with TEMPO in the oxidized (paramagnetic) and reduced (diamagnetic) states. The reduction was achieved by adding 5x molar access of sodium ascorbate[67]. The NMR spectra of polyUb:Ub4a$^{TEMPO}$ after reduction agreed with the spectra collected for polyUb:Ub4a, thus verifying that the presence of TEMPO did not adversely affect the structure of respective Ub species nor Ub4a peptide binding to polyUb. The PRE effects were quantified as the ratio of signal intensities ($I_{ox}/I_{red}$) in the $^1$H-$^{15}$N HSQC spectra of $^{15}$N-labeled Ub$_3$:Ub4a$^{TEMPO}$ with TEMPO in the oxidized ($I_{ox}$) and reduced ($I_{red}$) states[43]. The position of TEMPO's unpaired electron in the complex was reconstructed using the in-house Matlab program SLfit[68,69].

## Data availability

Atomic coordinates and structure factors have been deposited in the Protein Data Bank under accession codes 8F1F for Ub$_3$:Ub4a complex and 8E7O for Ub$_4$. NMR signal assignments for K48-linked Ub$_2$ used in this study are available under BMRB entry IDs 52127 and 52139, and for Ub$_1$ under BMRB entry ID 17769. Source data are provided with this paper.

## Code availability

The PRE data analysis was performed using Matlab program SLfit[68,69]; the code is available from Fushman Lab website [http://gandalf.umd.edu/FushmanLab/].

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

## Acknowledgements

This work was supported by NIH grant GM065334 to D.F. and A.B. NMR experiments were performed on instruments supported in part by NSF Grant DBI1040158. We acknowledge the use of beamline 22-ID of the Southeast Regional Collaborative Access Team (SER-CAT), located at the Advanced Photon Source, Argonne National Laboratory. Use of the Advanced Photon Source was supported by the U. S. Department of Energy, Office of Science, Office of Basic Energy Sciences, under Contract No. W-31-109-Eng-38.

## Author contributions

B.L. and B.G.W. synthesized all polyUb chains for this project and performed NMR studies; B.L. performed all biochemical assays and crystallized and determined the structure of $Ub_4$; J.L. and D.Z. crystallized the $Ub_3$:Ub4a complex, collected X-ray data and determined the structure; A.B. and G.B.V. synthesized the peptides; D.F. conceived the project, supervised the work, and contributed to NMR data analysis, H.S. supervised the peptide design, B.L., J.L, and D.F. wrote the manuscript with contributions from the other authors. All authors read and approved the final manuscript.

## Competing interests

The authors declare no competing interests
