## [Peer Review File · Nature Communications]

REVIEWER COMMENTS

Reviewer #1 (Remarks to the Author):

This is a well-written and highly informative manuscript that merges NMR, x-ray crystallography and enzymatic assays to define the mechanism of action for a macrocyclic peptide inhibitor of K48-linked Ub chains. The use of OTUB1 and IsoT to test and validate the NMR data is a strength of this manuscript as is the atomic level resolution structure obtained by x-ray crystallography and complementary NMR data in solution. This Reviewer is highly enthusiastic towards publication and has only minor comments/suggestions below.

In Supplementary Figure 2, the tetraUb chain retains a free state signal which is not present in the triUb experiment. Is the affinity stronger for triUb or is the peptide exchanging between different ubiquitins of the chain causing the proximal Ub in tetraUb to sometimes be free?

Given that the interactions involve the proximal free end of the chain how might the ligation to substrate influence Ub4a binding? The authors might want to comment on this point.

The figure referencing for Figure 2 at line 150 should be broken up in the sentence by panel as it was hard to find the data.

What is the band in Fig 4a/b at the top of the gel (just above 40 kDa)? This band should be labeled.

Reviewer #2 (Remarks to the Author):

The manuscript by Lemma et al. describes the structural and in part functional characterization of K48-linked polyubiquitin chains upon binding to a previously reported macrocyclic peptide-based ligand (ref 24 by some of the authors). Poly-ubiquitination plays a central role in proteasome-mediated protein degradation and moved into the focus of recent drug discovery efforts. Obtaining the different ubiquitin oligomers for biophysical characterization is challenging but can open the door to the identification of novel highly relevant inhibitors. To fully utilize such compounds, it is crucial to have a precise understanding of their binding mode. This however is a very rare for ubiquitin-targeting inhibitors. Herein the authors use state of the art biophysical techniques to address this question using both NMR and protein crystallography. In addition, the effect of inhibitor binding on ubiquitin chain assembly as well as degradation is studied. Overall, this is a very elegant and well performed study that contains a large amount of high-quality data. The findings herein can not only support the further development of related inhibitors but also pave the way to new design strategies. I recommend publication of this manuscript after the following minor points have been addressed:

- The abstract states 'n=4' that should be replaced e.g., by 'tetra-Ub'.
- The authors write: "The C-terminus of this Ub interacts closely with the side chain of R72 of UbC and with T12 of a symmetry-mate (this is a crystal-specific interaction)" a corresponding SI Figure should be added.

- The authors should compare the obtained structure of the 'Ub4a/Ub3' with unbound Ub structures (e.g. Ub4 obtained during their co-crystallization attempts). It would be interesting to see what differences occur.

Reviewer #3 (Remarks to the Author):

In this paper, Lemma et al have characterised the binding mode of a specific K48-polyubiquitin targeting cyclic peptide which they had previously discovered. In their previous work, they observed with NMR that the peptide specifically binds the proximal tri-ubiquitin of a free tetra-ubiquitin chain and that the C-terminal tail of the proximal ubiquitin is involved in this binding. Here, they confirmed the importance of the proximal C-terminal tail by truncating it and measuring ubiquitin chemical shifts upon peptide binding by NMR. They further used specific DUBs to confirm that the peptide specifically protected the proximal tri-ubiquitin when the tail is intact. They also show that the peptide prevents free K48 chain elongation beyond three ubiquitins. Finally, they determined the crystal structure of the cyclic peptide bound to tri-ubiquitin revealing how it is specifically recognized, and they validated this in solution using paramagnetic relaxation enhancement NMR.

In general, the experiments are performed well and the results convincing. The crystal structure showing how a small peptide can selectively bind a K48-linked tri-ubiquitin chain in particular is very exciting.

My main concern is that most of the manuscript concentrates on the role of the free C-terminus of a free chain, which would not exist when ubiquitin is conjugated onto a substrate. So, while this is interesting from a biophysical perspective, it is not as relevant for the activity of this peptide in cells which reduces the significance of this work. The authors argue that the Ub-R72-Ub3 mimics a real ubiquitinated substrate. This should be further investigated. For example, labelling the second-to-proximal ubiquitin in this case (UbR72-Ub*-Ub2) to see whether the C-terminal tail of conjugated ubiquitin makes similar interactions as free ubiquitin. Additionally the authors could specifically mutate residues in the tail of the second ubiquitin and measuring the affinity. It would also be interesting to see the affinities of the peptide for UbR72-Ub3 compared to wild-type Ub3, to see how much difference the free tail makes. With these experiments it would be easier to relate the biophysical findings to the activity of the peptide.

Additionally, the manuscript would be greatly improved if there was some specific exploration of which molecular interactions seen in the crystal structure/NMR are particularly important beyond the tail – i.e. mutating specific ubiquitin residues or peptide residues and measuring affinities. For example, the fact that the removal of the C-terminal tail completely stops binding to the proximal ubiquitin would suggest interactions with the proximal Ub hydrophobic patch have no importance.

Minor comments

1. Comparisons are made between figures 1, 2, S1 and S2, which to assess clearly requires opening multiple versions of the paper side-by-side.

2. Fig. 6a – from these viewpoints, it is difficult to see the colour of the actual interacting surface which would only be seen by slicing through the structure from the other direction.

3. In Fig 3f,g Ub4a addition is now on the left instead of the right as in the other panels, making it difficult to immediately follow.

Responses to reviewers

We would like to thank the reviewers for their valuable feedback and insights that helped us improve the manuscript. Below we provide point-by-point response to all reviewers' comments and detail how we modified our manuscript accordingly. Reviewers' comments are in black, while our responses are in blue.

Reviewer #1 (Remarks to the Author):

This is a well-written and highly informative manuscript that merges NMR, x-ray crystallography and enzymatic assays to define the mechanism of action for a macrocyclic peptide inhibitor of K48-linked Ub chains. The use of OTUB1 and IsoT to test and validate the NMR data is a strength of this manuscript as is the atomic level resolution structure obtained by x-ray crystallography and complementary NMR data in solution. This Reviewer is highly enthusiastic towards publication and has only minor comments/suggestions below.

In Supplementary Figure 2, the tetraUb chain retains a free state signal which is not present in the triUb experiment. Is the affinity stronger for triUb or is the peptide exchanging between different ubiquitins of the chain causing the proximal Ub in tetraUb to sometimes be free?

The peptide concentration in the tetra-Ub spectra shown in panel **c** of that figure did not reach complete saturation, resulting in the remaining signals of unbound tetra-Ub still present albeit significantly weaker in intensity. We used this figure also in order to highlight the fact that Ub4a binding to tetra-Ub is in slow-exchange regime on the NMR time scale, resulting in signals of both the peptide-bound and unbound species present in the spectra simultaneously and as distinct (un-averaged) signals. We thank the reviewer for reminding us about this, and added a sentence clarifying this point to the legend of Supplementary Figure 2.

Given that the interactions involve the proximal free end of the chain how might the ligation to substrate influence Ub4a binding? The authors might want to comment on this point.

In the Discussion section of the original manuscript, we suggested that tetra-Ub with fully deleted C-terminal tail mimics a tri-Ub anchored to a substrate (in this case, Ub_{R72}). In the revised manuscript we provide new experimental data that support this assumption and enabled us to address the question of whether conjugation to a substrate obstructs peptide's access to the C-terminal tail residues in polyUb (see also our response to Reviewer #3 comment). Specifically, we made two [Ub]₃-Ub_{R72} chains, one with ¹⁵N-labeled proximal Ub ([Ub]₃-¹⁵NUb_{R72}) and the other with ¹⁵N-labeled next-to-proximal Ub (Ub_B) unit ([Ub]₂-¹⁵NUb_B-Ub_{R72}) and used NMR to examine peptide's interactions with these chains. Our new results can be summarized as follows.

(1) The NMR signal perturbations in the Ub_B unit caused by Ub4a binding are very significant, and the Ub4a-bound spectra of that (next-to-proximal) Ub unit are essentially the same as of the

proximal Ub in un-truncated (wild-type) K48-linked tri-Ub. By contrast, only minimal signal perturbations were observed in the proximal, Ub_{R72} unit, except for the isopeptide-bonded side-chain NH group of K48. These results indicate that full deletion of the C-terminal tail of the proximal Ub in polyUb renders this unit (Ub_{R72}) essentially unrecognizable as 'ubiquitin' by the peptide, supporting our initial assumption that [Ub]₃-Ub_{R72} mimics tri-Ub conjugated to a model substrate (in this case, Ub_{R72}).

(2) The observed strong NMR signal shifts and attenuations indicate that the peptide has access to and interacts with both the C-terminal tail of Ub_B (conjugated to Ub_{R72}) and the side-chain NH group of K48 in Ub_{R72} involved in the isopeptide bond. This clearly indicates that conjugation to a substrate does not obstruct Ub4a binding to polyUb. Furthermore, in the course of titration with Ub4a, the NMR signals of Ub_B exhibited slow-exchange behavior (as in the proximal Ub of tri-Ub) consistent with strong binding.

These results are presented in a new section ("Insights into Ub4a binding to substrate-conjugated ubiquitin chains") and in new Figures 7 and Supplementary Figure 7, as well as the Discussion section. We believe that these new findings strengthened our manuscript, and would like to thank the reviewer for encouraging us to look into this.

We would also like to point out here (as mentioned in the Discussion section) that even if tethering to a substrate renders the entire C-terminal tail of the proximal Ub inaccessible, this does not abrogate Ub4a binding to long polyUb chains. In fact, our NMR and DUB assays for tetra-Ub with a fully deleted C-terminal tail demonstrate that for polyUb chains of length $n > 3$, required for efficient proteasomal targeting, obstructing peptide's access to the C-terminal tail of the proximal Ub will merely cause a shift of the binding mode/site towards the distal end of the chain with little effect on the strength of binding or the peptide's ability to prevent polyUb chain recognition by the proteasomal receptors or cleavage/disassembly by DUBs.

The figure referencing for Figure 2 at line 150 should be broken up in the sentence by panel as it was hard to find the data.

Done

What is the band in Fig 4a/b at the top of the gel (just above 40 kDa)? This band should be labeled.

That band corresponds to GST-fused E2-25K. It is now properly labeled in the Figure. Thank you for pointing out this omission.

Reviewer #2 (Remarks to the Author):

The manuscript by Lemma et al. describes the structural and in part functional characterization of K48-linked polyubiquitin chains upon binding to a previously reported macrocyclic peptide-based ligand (ref 24 by some of the authors). Poly-ubiquitination plays a central role in

proteasome-mediated protein degradation and moved into the focus of recent drug discovery efforts. Obtaining the different ubiquitin oligomers for biophysical characterization is challenging but can open the door to the identification of novel highly relevant inhibitors. To fully utilize such compounds, it is crucial to have a precise understanding of their binding mode. This however is a very rare for ubiquitin-targeting inhibitors. Herein the authors use state of the art biophysical techniques to address this question using both NMR and protein crystallography. In addition, the effect of inhibitor binding on ubiquitin chain assembly as well as degradation is studied. Overall, this is a very elegant and well performed study that contains a large amount of high-quality data. The findings herein can not only support the further development of related inhibitors but also pave the way to new design strategies. I recommend publication of this manuscript after the following minor points have been addressed:

- The abstract states 'n=4' that should be replaced e.g., by 'tetra-Ub'.

Done

- The authors write: "The C-terminus of this Ub interacts closely with the side chain of R72 of UbC and with T12 of a symmetry-mate (this is a crystal-specific interaction)" a corresponding SI Figure should be added.

Following the reviewer's request, we added panel c illustrating these contacts to Supplementary Figure 4, updated the Figure legend, and inserted a reference to this panel in the text.

- The authors should compare the obtained structure of the 'Ub4a/Ub3' with unbound Ub structures (e.g. Ub4 obtained during their co-crystallization attempts). It would be interesting to see what differences occur.

We added Supplementary Figure 9 showing such a comparison and referred to in the text.

Reviewer #3 (Remarks to the Author):

In this paper, Lemma et al have characterised the binding mode of a specific K48-polyubiquitin targeting cyclic peptide which they had previously discovered. In their previous work, they observed with NMR that the peptide specifically binds the proximal tri-ubiquitin of a free tetra-ubiquitin chain and that the C-terminal tail of the proximal ubiquitin is involved in this binding. Here, they confirmed the importance of the proximal C-terminal tail by truncating it and measuring ubiquitin chemical shifts upon peptide binding by NMR. They further used specific DUBs to confirm that the peptide specifically protected the proximal tri-ubiquitin when the tail is intact. They also show that the peptide prevents free K48 chain elongation beyond three ubiquitins. Finally, they determined the crystal structure of the cyclic peptide bound to tri-ubiquitin revealing how it is specifically recognized, and they validated this in solution using paramagnetic relaxation enhancement NMR.

In general, the experiments are performed well and the results convincing. The crystal structure

showing how a small peptide can selectively bind a K48-linked tri-ubiquitin chain in particular is very exciting.

My main concern is that most of the manuscript concentrates on the role of the free C-terminus of a free chain, which would not exist when ubiquitin is conjugated onto a substrate. So, while this is interesting from a biophysical perspective, it is not as relevant for the activity of this peptide in cells which reduces the significance of this work. The authors argue that the Ub-R72-Ub3 mimics a real ubiquitinated substrate. This should be further investigated. For example, labelling the second-to-proximal ubiquitin in this case (UbR72-Ub*-Ub2) to see whether the C-terminal tail of conjugated ubiquitin makes similar interactions as free ubiquitin. Additionally the authors could specifically mutate residues in the tail of the second ubiquitin and measuring the affinity. It would also be interesting to see the affinities of the peptide for UbR72-Ub3 compared to wild-type Ub3, to see how much difference the free tail makes. With these experiments it would be easier to relate the biophysical findings to the activity of the peptide.

First, we would like to point out that free/unanchored polyubiquitin chains, including K48-linked chains, do exist in cells, and there is mounting evidence that they play important physiological roles (recently reviewed in Blount, J. R., Johnson, S. L., and Todi, S. V., Unanchored Ubiquitin Chains, Revisited, *Front Cell Dev Biol* 8, 582361 (2020)). Thus, our results have direct relevance to these polyUb chains.

Second, we thank the reviewer for encouraging us to closely examine Ub4a binding to the second-to-proximal Ub. In fact, following the reviewer's suggestions, we made two [Ub]₃-Ub_{R72} chains, one with ¹⁵N-labeled proximal Ub ([Ub]₃-¹⁵NUb_{R72}) and one with ¹⁵N-labeled next-to-proximal, Ub_B unit ([Ub]₂-¹⁵NUb-Ub_{R72}), and used NMR to examine peptide's interactions with these chains. Our new results (spelled out in greater detail in the revised ms and above in our response to reviewer #1 comment) can be summarized as follows.

(1) Our NMR data (presented in Figures 7 and S7) indicate that full deletion of the C-terminal tail of the proximal Ub in polyUb renders this unit (Ub_{R72}) essentially unrecognizable as 'ubiquitin' by the Ub4a peptide, thus supporting our initial assumption that [Ub]₃-Ub_{R72} mimics a tri-Ub conjugated to a model substrate (i.e., Ub_{R72}).

(2) Our NMR data (Figures 7e and S7) indicate that Ub4a interactions with the next-to-proximal Ub (Ub_B) in tetra-Ub with fully deleted C-terminal tail are essentially the same as with the proximal Ub in the un-truncated (wild type) tri-Ub. Furthermore, our data revealed that the peptide has access to and interacts with both the C-terminal tail of Ub_B (conjugated to Ub_{R72}) and the side-chain NH group of K48 in Ub_{R72} involved in the isopeptide bond. Furthermore, in the course of titration with Ub4a, the NMR signals of Ub_B exhibited slow-exchange behavior (as in the proximal Ub of wild-type tri-Ub) consistent with strong binding.

We believe that these results support the relevance of our findings to both unanchored and substrate-conjugated Ub chains.

We did not perform mutations of individual residues in the C-terminal tail. We believe that these studies are beyond the scope of the current paper, and we plan do them in the future. We would like to mention in this regard that mutations in the C-terminal tail can affect the efficiency of ubiquitin activation by the E1 enzyme, reducing the ability to efficiently generate polyUb chains containing such mutations (see, for example, Singh et al., The Hydrophobic Patch of Ubiquitin is Important for its Optimal Activation by Ubiquitin Activating Enzyme E1, *Anal Chem* 89, 7852 (2017)).

Additionally, the manuscript would be greatly improved if there was some specific exploration of which molecular interactions seen in the crystal structure/NMR are particularly important beyond the tail – i.e. mutating specific ubiquitin residues or peptide residues and measuring affinities. For example, the fact that the removal of the C-terminal tail completely stops binding to the proximal ubiquitin would suggest interactions with the proximal Ub hydrophobic patch have no importance.

We thank the reviewer for these suggestions that bring up interesting questions. We believe that it is a combination of peptide's interactions simultaneously with the hydrophobic patch residues and the tail (possibly including the isopeptide linkage) that together contribute to the strong binding, likely through avidity effect. The peptide's interactions with the tail certainly play an important role, but because the tail is flexible (as we showed previously, see Castañeda et al., Linkage-specific conformational ensembles of non-canonical polyubiquitin chains, *Phys Chem Chem Phys* 18, 5771 (2016)), it is perhaps peptide's interactions with the hydrophobic patch residues that properly position the peptide in the proximity of the tail residues to enable/facilitate its interactions with the C-terminal tail.

Deconvolution of contributions to peptide binding from the tail and the hydrophobic patch residues in ubiquitin requires a detailed and thorough investigation, which is beyond the scope of this study focused primarily on the large-scale structural rearrangements accompanying the peptide binding. We intend to pursue these studies in the future. In this regard we would like to point out that because ubiquitin's hydrophobic patch residues play important roles in ubiquitin recognition by the E1 and E2 enzymes, mutating these residues creates an additional complication to the ability to efficiently generate Ub chains containing such mutations (see, for example, Singh et al., The Hydrophobic Patch of Ubiquitin is Important for its Optimal Activation by Ubiquitin Activating Enzyme E1, *Anal Chem* 89, 7852 (2017)). Furthermore, noncovalent intra-chain interactions between Ub units in K48-linked polyUb are mediated by contacts between their hydrophobic patch residues (see, for example, our structure of tetraUb in Supplementary Figure 8). Thus, mutating Ub hydrophobic patch residues will weaken or even disrupt those interactions, inadvertently affecting the change in free energy (and the K_d) associated with tetraUb:peptide binding. Thus, such studies might not be as straightforward as it might seem.

On a qualitative level, our observations that after full deletion of the C-terminal tail of the proximal Ub, the next-to-proximal Ub unit (Ub_B) exhibits slow-exchange binding (as illustrated in Figure 7a of the revised ms) and its spectra in the Ub4a-bound state are almost indistinguishable from those of the Ub4a-bound proximal Ub in the un-truncated tri-Ub chain (Figures 7e and S7c), suggest that the peptide binding is still strong.

Minor comments

1. Comparisons are made between figures 1, 2, S1 and S2, which to assess clearly requires opening multiple versions of the paper side-by-side.

We understand this issue. However, we find it impossible to squeeze all these images into a single figure. Besides, even a comparison of two figures from the main text simultaneously requires opening two electronic versions of the paper, side by side.

2. Fig. 6a – from these viewpoints, it is difficult to see the colour of the actual interacting surface which would only be seen by slicing through the structure from the other direction.

The main purpose of coloring the residues exhibiting NMR signal perturbations was to illustrate/emphasize that those residues are located at the ubiquitin-peptide interface observed in the crystal structure. We did not intend to reveal the minute details of these contacts recognizing the fact that NMR signal perturbations merely reflect changes in the electronic environment of a nucleus, and it is not always possible to directly pinpoint a particular interatomic interaction/contact based on chemical shift perturbations or signal attenuations. However, following the reviewer's suggestion, we included in panel **a** of Figure 6 a slice view of the inside of the central hole of the complex (including the peptide).

3. In Fig 3f,g Ub4a addition is now on the left instead of the right as in the other panels, making it difficult to immediately follow.

Following the reviewer's suggestion, we broke the gels in panels **f** and **g** of Figure 3 into two sub-panels and rearranged their order to be consistent with the other gels in that figure. We hope this made these gels easier to follow.

In addition, we fixed several typos in the text and figures.

All the changes in the manuscript and SI texts are marked using blue font.

Addressing comments and suggestions of reviewers #2 and #3 required additional biochemical work and NMR studies. These were performed by Bryan G. Wentz who is now included as a co-author of our revised manuscript.

REVIEWERS' COMMENTS

Reviewer #3 (Remarks to the Author):

The authors have provided new experiments which have addressed my previous concerns. I now fully support publication of this manuscript.